# Characterizing and Optimizing the Spatial Kernel of Multi Resolution Hash Encodings

**Tianxiang Dai & Jonathan Fan**[*]
Department of Electrical Engineering
Stanford University
Stanford, CA 94305, USA
{txdai, jonfan}@stanford.edu

## Abstract

Multi-Resolution Hash Encoding (MHE), the foundational technique behind Instant Neural Graphics Primitives, provides a powerful parameterization for neural fields. However, its spatial behavior lacks rigorous understanding from a physical systems perspective, leading to reliance on heuristics for hyperparameter selection. This work introduces a novel analytical approach that characterizes MHE by examining its Point Spread Function (PSF), which is analogous to the Green's function of the system. This methodology enables a quantification of the encoding's spatial resolution and fidelity. We derive a closed-form approximation for the collision-free PSF, uncovering inherent grid-induced anisotropy and a logarithmic spatial profile. We establish that the idealized spatial bandwidth, specifically the Full Width at Half Maximum (FWHM), is determined by the average resolution, $N_{\mathrm{avg}}$. This leads to a counterintuitive finding: the effective resolution of the model is governed by the broadened empirical FWHM (and therefore $N_{\mathrm{avg}}$), rather than the finest resolution $N_{\mathrm{max}}$, a broadening effect we demonstrate arises from optimization dynamics. Furthermore, we analyze the impact of finite hash capacity, demonstrating how collisions introduce speckle noise and degrade the Signal-to-Noise Ratio (SNR). Leveraging these theoretical insights, we propose Rotated MHE (R-MHE), an architecture that applies distinct rotations to the input coordinates at each resolution level. R-MHE mitigates anisotropy while maintaining the efficiency and parameter count of the original MHE. This study establishes a methodology based on physical principles that moves beyond heuristics to characterize and optimize MHE.

## 1 Introduction

Multi-Resolution Hash Encoding (MHE) (Müller et al., 2022), the central innovation underlying Instant Neural Graphics Primitives (Instant-NGP), has catalyzed significant advancements in neural fields, enabling accelerated optimization and real-time rendering for applications such as Neural Radiance Fields (NeRF) (Mildenhall et al., 2020) and Signed Distance Functions (SDFs). This even extends beyond computer graphics and was applied to PINNs (Huang & Alkhalifah, 2024) and physical designs (Dai et al., 2025). MHE offers a compact and efficient parameterization; however, its behavior is critically dependent on hyperparameters, including the number of levels $L$, growth factor $b$, resolutions $(N_{\mathrm{max}}, N_{\mathrm{min}})$, and hash table capacity $T$. These parameters are typically selected using generalized heuristics. Despite extensive research into neural field architectures (Sun et al., 2022; Chen et al., 2022; Fridovich-Keil et al., 2023) and anti-aliasing techniques (Greer et al., 2021), a substantial gap persists: the study of MHE, and NeRF models more generally, currently lacks rigorous analysis from a physical systems perspective.

In this work, we introduce a novel methodology to characterize and understand the performance of MHE by analyzing its *Point Spread Function* (PSF). Analogous to measuring the Green's function of a physical system, the PSF characterizes the model's response when optimized to represent an idealized point source (Figure 1b). This approach permits the rigorous quantification of effective

---

[*]Corresponding author.

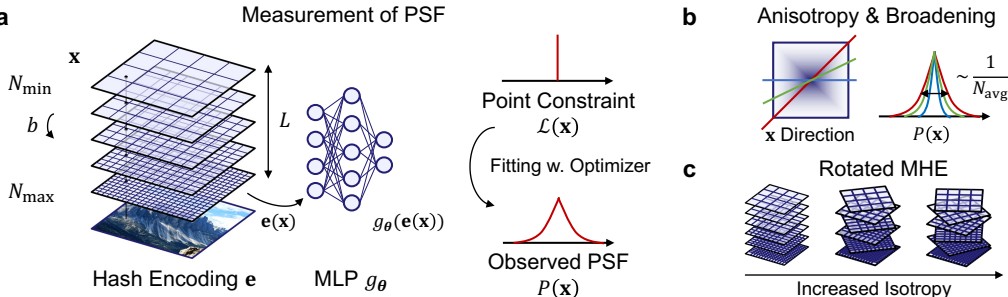

Figure 1: **Overview of MHE Characterization and Optimization.** **(a)** The MHE architecture utilizes $L$ grid levels with resolutions growing by a factor $b$. The encoding $\mathbf{e}(\mathbf{x})$ is passed to an MLP $g_{\boldsymbol{\theta}}$. We characterize the system by optimizing for a point constraint and measuring the resulting Point Spread Function (PSF). **(b)** This analysis reveals inherent grid-induced anisotropy (narrower along axes) and optimization-induced broadening, establishing that the effective resolution (FWHM) scales with $1/N_{\mathrm{avg}}$. **(c)** To mitigate anisotropy, we propose Rotated MHE (R-MHE), which applies distinct rotations at each resolution level, leading to a more isotropic PSF.

spatial resolution and the identification of performance issues that often contradict intuition derived from the architecture's specifications. We isolate the encoding by operating in a linearized decoder regime, a framework motivated by kernel perspectives (Jacot et al., 2018) and spectral analysis (Tancik et al., 2020).

Our analysis begins with the examination of the PSF of the idealized, collision free MHE. We derive a closed form approximation demonstrating that the PSF exhibits logarithmic radial decay and significant *grid induced anisotropy*, inherited from the underlying interpolation kernels (Keys, 1981). Theoretically, the idealized Full Width at Half Maximum (FWHM) is determined by the average resolution, $N_{\mathrm{avg}}$.

We confirm these trends through numerical experiments, which reveal that optimization dynamics induce significant spatial broadening compared to the idealized minimum norm prediction. This confirms that the effective two point resolution of the model is substantially lower then $N_{\mathrm{max}}$ and governed by the broadened empirical FWHM (and thus $N_{\mathrm{avg}}$).

We further investigate the impact of finite hash capacity, demonstrating how collisions introduce speckle-like side lobes and degrade the Signal-to-Noise Ratio (SNR). Informed by our comprehensive PSF analysis, we demonstrate how these insights can be leveraged to improve reconstruction quality. We introduce *Rotated MHE (R-MHE)* (Figure 1c), an architecture that applies distinct rotations to the input coordinates at each resolution level. By utilizing the existing multi-resolution structure, R-MHE improves isotropy without requiring additional hash tables or parameters, maintaining the efficiency of the original MHE.

**Contributions.** This work establishes a new framework based on physical principles for analyzing MHE, providing several key advancements:

- We derive a closed-form approximation for the MHE Point Spread Function, rigorously characterizing its anisotropic and logarithmic spatial profile, and identifying the average resolution, $N_{\mathrm{avg}}$, as the principal determinant of the idealized FWHM.

- We reveal and characterize optimization-induced spatial broadening, demonstrating theoretically and empirically that it arises from spectral bias.

- We provide an evaluation of the impact of hash collisions on the Signal-to-Noise Ratio (SNR).

- We introduce Rotated MHE (R-MHE), a novel, parameter-free modification that improves isotropy by applying distinct rotations at each resolution level.

- We validate a principled methodology guided by the PSF analysis for hyperparameter selection that demonstrably outperforms standard heuristics.

## 2 BACKGROUND AND PRELIMINARIES

### 2.1 RELATED WORK

Our analysis draws upon and contributes to several interconnected areas of research.

**Neural Fields, Encodings, and Spectral Analysis.** The introduction of NeRF established coordinate-based volumetric rendering (Mildenhall et al., 2020). Positional encodings, including Fourier features (Tancik et al., 2020) and periodic activations (Sitzmann et al., 2020), are known to shape optimization dynamics and frequency bias (Rahaman et al., 2018). Adopting a Neural Tangent Kernel (NTK) viewpoint, where linearized training dynamics justify the analysis of an encoding's induced kernel (Jacot et al., 2018), we utilize this theoretical lens to derive an explicit PSF for MHE (Müller et al., 2022).

**Explicit Grids and Factorized Structures.** Researchers have explored replacing or augmenting MLP decoders with explicit spatial representations, including voxel grids (Sun et al., 2022; Peng et al., 2022), tensor factorizations (Chen et al., 2022), and planar factorizations (Fridovich-Keil et al., 2023). While MHE is widely adopted, extensions such as Dictionary Fields (Chen et al., 2023) have been proposed to improve expressivity. Our work is complementary; R-MHE improves the underlying grid structure and could potentially be integrated with these extensions. We aim to elucidate *how* MHE behaves spatially and provide principles for its optimization.

**Interpolation Kernels and Anisotropy.** The separable tent kernel underlying multilinear interpolation (Keys, 1981; Thevenaz et al., 2000) inherently induces differences in effective blur along axes versus diagonals. We demonstrate that MHE inherits these anisotropies across multiple scales, resulting in direction-dependent FWHM and resolution limits even without hash collisions.

**Hashing and Collisions.** Spatial hashing has a long history in computer graphics (Lefebvre & Hoppe, 2006) and real-time reconstruction (Nießner et al., 2013). Our collision analysis formalizes how finite capacity hash tables translate into PSF speckle and SNR loss within MHE.

### 2.2 MHE ARCHITECTURE REVIEW

The MHE aims to learn a function $f(\mathbf{x}) = g_{\boldsymbol{\theta}}(\mathbf{e}(\mathbf{x}))$. It utilizes $L$ resolution levels defined by $N_l = N_{\min} \cdot b^l$. At each level $l$, features are retrieved from a table $\mathbf{F}^l$ of size $T$ using a spatial hash function $\mathcal{H}$ and multilinear interpolation. The interpolation kernel $K(\mathbf{u})$ is constructed as the product of 1D tent functions: $K(\mathbf{u}) = \prod_{d=1}^{D} \max(0, 1 - |u_d|)$. Consequently, the spatial kernel at level $l$ is given by $K_l(\mathbf{x}) = K(N_l \mathbf{x})$.

The idealized spatial response of the encoding, averaged over all possible grid alignments, is characterized by the induced kernel $B_l(\mathbf{x})$, which is the auto-correlation of the interpolation kernel: $B_l(\mathbf{x}) = (K_l * K_l)(\mathbf{x})$. This results in a separable cubic B-spline kernel (Thevenaz et al., 2000).

## 3 CHARACTERIZING THE MHE SPATIAL KERNEL

We analyze the Point Spread Function (PSF) to characterize the intrinsic spatial behavior of the MHE architecture, examining the system's response when optimized under sparse constraints. To isolate the properties of the encoding from the influence of the subsequent MLP decoder $g_{\boldsymbol{\theta}}$, we assume the MLP can be approximated by its linearization, $f(\mathbf{x}) \approx \mathbf{W}\mathbf{e}(\mathbf{x})$. This is justified by experiments regarding MLP depth (Appendix D.4).

### 3.1 THE IDEALIZED, COLLISION FREE PSF

We first consider the optimization process for a single point constraint $\mathcal{L} = (f(\mathbf{x}_0) - A)^2$. In this idealized analysis, we assume the absence of hash collisions (infinite $T$). Under the linearized framework and the minimum norm assumption, the responsibility is distributed equally across all $L$ levels. The resulting idealized PSF $P_{\text{Ideal}}(\mathbf{x})$ is the average superposition of the normalized induced

kernels $\hat{B}_l(\mathbf{x})$ (the cubic B-spline, Section 2.2):

$$P_{\text{Ideal}}(\mathbf{x}) = \frac{1}{L} \sum_{l=0}^{L-1} \hat{B}_l(\mathbf{x} - \mathbf{x}_0) \tag{1}$$

**Generalized Closed-Form Approximation and Anisotropy.** The induced kernel $\hat{B}(\mathbf{x})$ (cubic B-spline) is separable but inherently anisotropic. We derive a generalized closed-form approximation by approximating the summation with an integral, utilizing the Taylor expansion of the B-spline kernel near the center (Appendix A.3).

Let $\mathbf{v} = N_{\min}\mathbf{x}$ be the normalized position. The closed-form approximation near the center can be expressed as:

$$P_{\text{Ideal}}(\mathbf{v}) \approx \frac{1}{L \ln(b)} \left[ -\ln(\|\mathbf{v}\|) + C_D - A_D(\mathbf{v}) + O(\mathbf{v}^2) \right] \tag{2}$$

This expression reveals a dominant **logarithmic decay profile** ($-\ln(\|\mathbf{v}\|)$) modulated by an anisotropy factor $A_D(\mathbf{v})$ specific to the B-spline kernel. As proven in Appendix A.3, the B-spline kernel is narrower along the axes.

**FWHM and Average Resolution.** The Full Width at Half Maximum (FWHM) is direction-dependent. We define the inherent broadening factor of the idealized induced kernel as $\beta_{\text{ideal}}$. The FWHM of the 1D cubic B-spline kernel is numerically calculated to be $\beta_{\text{ideal}} \approx 1.18$. The FWHM of the composite PSF along the axes scales proportionally to the average resolution $N_{\text{avg}}$ (Appendix A.3):

$$\Delta_{\text{Axis, Ideal}} \approx \beta_{\text{ideal}}/N_{\text{avg}} \approx 1.18/N_{\text{avg}} \tag{3}$$

The idealized spatial bandwidth is dictated by $N_{\text{avg}}$, while the FWHM along the diagonals is comparatively wider.

## 3.2 Empirical Validation and Optimization-Induced Broadening

We validate these theoretical results using a customized 2D implementation based on the Instant-NGP framework. We configure MHE networks (varying $L$ and $b$) with a sufficiently large $T$ to minimize collisions, and optimize for a single point objective $\mathcal{L} = (f(\mathbf{0}) - 1)^2$.

Empirically, we observe that the realized PSF, $P_{\text{Empirical}}(\mathbf{x})$, is significantly broader than the idealized minimum-norm prediction $P_{\text{Ideal}}(\mathbf{x})$. We characterize this additional broadening by introducing an optimization-induced spatial broadening factor $\beta_{\text{opt}}$, such that $P_{\text{Empirical}}(\mathbf{x}) \approx P_{\text{Ideal}}(\mathbf{x}/\beta_{\text{opt}})$. Because the idealized B-spline model already accounts for the inherent anisotropy, the optimization-induced broadening $\beta_{\text{opt}}$ can be accurately modeled as isotropic.

We define the total empirical broadening factor $\beta_{\text{emp}}$ such that the empirical FWHM along the axis is $\Delta_{\text{Axis, Emp}} = \beta_{\text{emp}}/N_{\text{avg}}$. This combines the idealized broadening and the optimization-induced broadening:

$$\beta_{\text{emp}} = \beta_{\text{ideal}} \cdot \beta_{\text{opt}} \tag{4}$$

**Spectral Bias.** This optimization-induced broadening ($\beta_{\text{opt}} > 1$) occurs because the optimization process (e.g., using Adam) does not converge to the minimum-norm solution. Gradient-based optimization exhibits implicit biases, often referred to as spectral bias (Rahaman et al., 2018), where lower frequencies are learned preferentially. This leads the optimization trajectory to a solution where coarse features (low $N_l$) are prioritized over fine features. This effective re-weighting towards lower frequencies results in the observed spatial broadening. We provide a theoretical derivation in Appendix D, modeling the weights as $w_l \propto (N_l)^{-\gamma}$, where $\gamma$ is the spectral bias exponent. We prove that $\beta_{\text{opt}}$ increases monotonically with $\gamma$, and argue that $\gamma$ (and thus $\beta_{\text{opt}}$) increases with the spatial dimension $D$.

**Characterizing the Broadening Factors.** We consistently observe a total empirical broadening of $\beta_{\text{emp}} \approx 3.0$ across various configurations of $L$ and $b$ when using the Adam optimizer (Figure 2). To understand the sensitivity of $\beta_{\text{opt}}$, we conducted experiments varying the optimizer, MLP architecture, and training dynamics (Appendix D.4, D.5). We found that $\beta_{\text{opt}}$ is primarily dependent

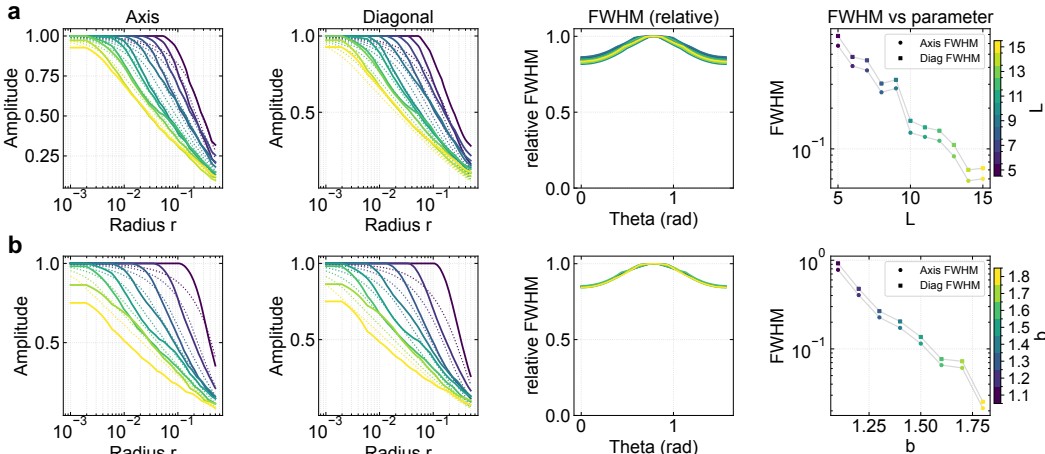

Figure 2: **Numerical Validation of the MHE PSF (2D).** We analyze the empirical PSF (solid lines) compared to the broadened theoretical prediction (dotted lines, incorporating the total empirical broadening $\beta_{\text{emp}} \approx 3.0$). **(a)** Varying $L$ (fixed $b = 1.5$). **(b)** Varying $b$ (fixed $L = 10$). **(Columns 1 & 2)** Cross-sections along the Axis and Diagonal show characteristic anisotropy (broader on diagonal). The broadened theory accurately matches the empirical decay. **(Column 3)** Relative FWHM vs. angle confirms the B-spline anisotropy (narrower along axes, $\theta = 0$). **(Column 4)** The empirical FWHM aligns well with the theoretical trends dictated by $N_{\text{avg}}$. Colors indicate the varied parameter ($L$ or $b$).

on the optimizer choice (e.g., corresponding to $\beta_{\text{emp}} \approx 3.0$ for Adam variants in 2D and 3D, Fig. 10), but is remarkably stable across hyperparameters and insensitive to the MLP depth. This stability confirms that $\beta_{\text{opt}}$ is a robust characteristic of the optimization dynamics for a given setup.

Figure 2 compares the empirical PSF with the theoretical predictions. We observe excellent agreement between the empirical results (solid lines) and the broadened theory (dotted lines, incorporating $\beta_{\text{emp}} \approx 3.0$). The characteristic anisotropy (broader on diagonals) predicted by the B-spline model is clearly visible (Columns 1-3). Furthermore, the empirical decay confirms the logarithmic profile predicted by the closed-form approximation (Eq. 2), and the scaling of the FWHM with respect to $L$ and $b$ aligns with the theoretical trends (Column 4).

### 3.3 RESOLUTION LIMITS AND TWO POINT INTERACTIONS

We analyze the system's behavior when optimized for two closely spaced point constraints, $\mathbf{x}_A$ and $\mathbf{x}_B$, separated by distance $d$. Under the linearized assumption, the reconstruction $R(\mathbf{x})$ is the superposition of the individual PSFs.

**Constructive Interference: Idealized vs. Empirical Resolution.** The resolution limit $d_{\text{crit}}$ (Rayleigh criterion) is the minimum distance such that a dip exists between two peaks. In the idealized minimum-norm configuration (Appendix A.5), the smoothness of the B-spline kernel implies the theoretical resolution limit is infinitesimal. However, our empirical analysis reveals that the practical resolution limit is dictated by the broadened empirical PSF (Section 3.2). The ability to resolve two points is therefore governed by the empirical FWHM, which scales with $1/N_{\text{avg}}$.

**Destructive Interference and the Dipole Response.** For a spatial dipole, $R(\mathbf{x}) \approx P(\mathbf{x} - \mathbf{x}_A) - P(\mathbf{x} - \mathbf{x}_B)$. When $d$ is small, $R(\mathbf{x}) \approx d \cdot \nabla P(\mathbf{x})$. We find that the spatial behavior and extent of the dipole response are characterized by the empirical FWHM of the underlying PSF, due to the optimization induced broadening. When the separation is smaller than FWHM, the maximum values no longer appear at the points and artifacts appear.

**Numerical Validation: Two-Point Interactions.** We extend the 2D experimental setup, optimizing for two point constraints while varying MHE parameters and separation $d$. For constructive

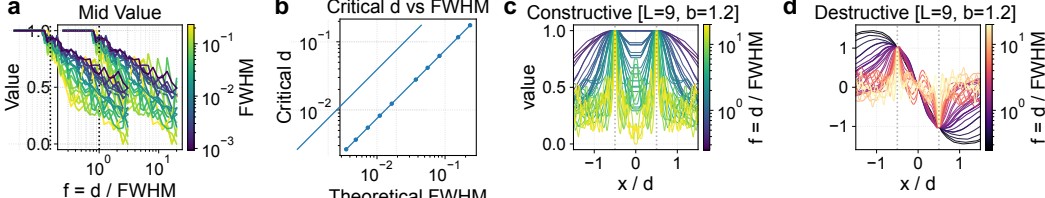

Figure 3: **Empirical Analysis of Two-Point Interactions (2D).** We analyze resolution by optimizing for two nearby points, normalizing separation by $f = d/\text{FWHM}$. **(a)** The midpoint value between two constructive peaks drops significantly when $d \approx \text{FWHM}$. **(b)** The empirically measured critical distance $d_{\text{crit}}$ scales linearly with the FWHM across various MHE configurations, confirming that FWHM ($N_{\text{avg}}$) dictates the practical resolution limit. **(c, d)** Constructive and destructive (dipole) interference profiles. The consistent shape confirms that the FWHM characterizes the spatial response.

interference (Figure 3(a)), a significant dip emerges when the separation is approximately equal to the empirical FWHM. We empirically determine the critical distance $d_{\text{crit}}$ across various configurations and find a direct linear relationship with the FWHM (Figure 3(b)). This confirms that the practical two-point resolution limit is determined by the empirical FWHM (scaling with $1/N_{\text{avg}}$), not $N_{\text{max}}$. For the dipole configuration (Figure 3(d)), the profiles exhibit a consistent shape characterized by the FWHM, illustrating that the spatial extent of the dipole response is also governed by $N_{\text{avg}}$.

## 4 THE IMPACT OF FINITE HASH CAPACITY

Having characterized the inherent spatial properties of the MHE kernel, we now analyze the effects of practical memory constraints. In real implementations, the hash table capacity $T$ is finite, leading to collisions where different spatial vertices map to the same entry in the feature table $\mathbf{F}^l$.

### 4.1 MODELING HASH COLLISIONS AND SPECKLE

When vertices collide, they share the same feature vector. For a single point objective at $\mathbf{x}_0$, the optimized feature vector $\mathbf{F}_i^l$ becomes proportional to the sum of the interpolation weights of all colliding vertices $C_i^l$ evaluated at the target point:

$$\mathbf{F}_i^l \propto \sum_{\mathbf{v} \in C_i^l} K_l(\mathbf{v} - \mathbf{x}_0) \tag{5}$$

A vertex $\mathbf{v}$ near $\mathbf{x}_0$ might collide with a spatially distant vertex $\mathbf{v}'$. The optimized feature is now inadvertently activated when querying $\mathbf{v}'$. This mechanism causes unintended "ghost" responses far from the center of the PSF, resulting in spurious side lobes or speckle patterns in the spatial response. We model the resulting PSF as $P_{\text{Collision}}(\mathbf{x}) = P_{\text{Ideal}}(\mathbf{x}) + n(\mathbf{x})$. The severity of the noise $n(\mathbf{x})$ depends strongly on the *collision ratio* (load factor). As the collision ratio increases, the variance of $n(\mathbf{x})$ increases, and the Signal to Noise Ratio (SNR) decreases.

### 4.2 NUMERICAL VALIDATION OF COLLISION EFFECTS

We investigate the impact of collisions experimentally by training a 2D MHE network with a single point objective while systematically varying $T$, $L$, and $b$. Figure 4 summarizes the impact of these parameters on the SNR. Across all configurations, increasing the collision ratio eventually leads to a significant degradation in SNR. We observe that increasing $L$ (Panel a) or $b$ (Panel b) generally improves the achievable SNR for a fixed capacity $T$. This suggests that distributing the representation across more levels or with greater separation enhances robustness, provided that $T$ is sufficient.

## 5 ROTATED MHE (R-MHE)

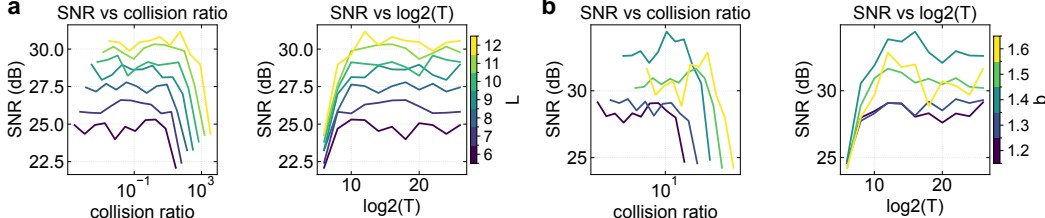

Figure 4: **Quantitative Analysis of Collision Effects on SNR (2D).** We analyze the SNR of the empirical PSF as a function of collision ratio and hash table size $T$. **(a)** Impact of varying the number of levels $L$ (fixed $b = 1.5$). **(b)** Impact of varying the growth factor $b$ (fixed $L = 10$). In all cases, SNR degrades rapidly at high collision ratios (low $T$). Higher $L$ or $b$ generally improves the achievable SNR for a fixed $T$. Colors indicate the varied parameter.

Our analysis identified that the reliance on axis-aligned grids in standard MHE leads to inherent anisotropy (Section 3.1). This is undesirable in applications like NeRF where viewing angles vary continuously, or in image regression where features may not align with the axes. To address this limitation, we propose the Rotated MHE (R-MHE) architecture.

## 5.1 MOTIVATION AND ARCHITECTURE

R-MHE leverages the existing multi-resolution structure of MHE. Instead of using a single rotation for the entire encoding or requiring multiple independent hash tables, R-MHE applies a distinct rotation matrix $\mathbf{R}_l$ to the input coordinates $\mathbf{x}$ specifically at each resolution level $l$. The encoding process at level $l$ is modified as follows:

$$\mathbf{e}_l(\mathbf{x}) = \text{Interpolate}(\mathbf{F}^l, \mathcal{H}(\lfloor N_l \mathbf{R}_l \mathbf{x} \rceil)) \tag{6}$$

This model utilizes the same hash function $\mathcal{H}$ and feature tables $\mathbf{F}^l$ as standard MHE, maintaining the exact memory footprint, parameter count, and computational efficiency.

## 5.2 ROTATION STRATEGIES

The key to R-MHE is selecting a set of rotations $\{\mathbf{R}_l\}$ that maximizes the diversity of grid orientations across the levels. In 2D, we employ a progressive rotation strategy. We define a base rotation angle $\theta$, and set the rotation at level $l$ to be $\mathbf{R}_l = \text{Rot}(l \cdot \theta)$. This ensures that subsequent levels are oriented differently, maximizing the angular coverage over the $L$ levels. We analyze the impact of the choice of $\theta$ in Section 5.4. In 3D, we aim for uniform sampling of the rotation space SO(3). We utilize the vertex orientations of regular polyhedra (tetrahedron, cube, octahedron, icosahedron) to define a set of canonical directions. The rotations $\{\mathbf{R}_l\}$ are chosen to align the grid axes with these directions, cycle through the vertices of one chosen polyhedron type across the levels $L$ (Details in Appendix A.6).

## 5.3 THEORETICAL BENEFITS: IMPROVED ISOTROPY

The R-MHE architecture offers significant advantage of improved isotropy derived directly from our PSF analysis.

The resulting idealized PSF of R-MHE is the superposition of the rotated induced kernels:

$$P_{\text{R-MHE}}(\mathbf{x}) = \sum_{l=0}^{L-1} \frac{1}{L} \hat{B}_l(\mathbf{R}_l(\mathbf{x} - \mathbf{x}_0)) \tag{7}$$

where the rotations are applied at each level. This averaging process across differently oriented levels effectively mitigates the angular dependencies inherent in the standard MHE, leading to a more uniform spatial resolution across different orientations.

## 5.4 EXPERIMENTAL VALIDATION OF R-MHE

We validate the benefits of the R-MHE architecture using the established numerical framework and a practical 2D application. We analyze the impact of the base rotation angle $\theta$. We define the strategy by the parameter M, such that $\theta = 90 \deg /M$. $M$ represents the effective number of unique orientations sampled within the first quadrant.

**Quantifying Isotropy.** We conducted experiments to measure the PSF for standard MHE ($M = 1$) and R-MHE with increasing $M$ (up to 16). The objective was to quantify the improvement in isotropy (measured by the Anisotropy Ratio, the ratio of the maximum to minimum distance to center across directions at different levels). Analysis of this ratio and the kernels in 5 (a) and (b) reveals that for this averaging contibutes to better isotropy for moderate $M$. This trend suggests that while increasing M improves isotropy by diversifying orientations, excessively large M would cause L levels to be insufficient to effectively average a very large number of unique orientations, potentially reducing the effectiveness of the rotation strategy. From $M = 1$ in 5 (a) it is evident that the anisotropy ratio of 1.17 proven in A.4 holds across many different levels.

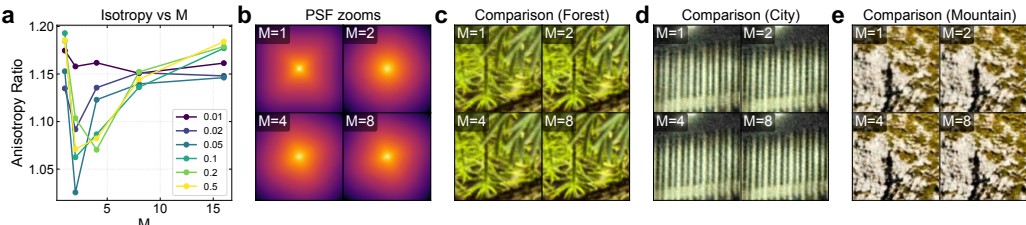

Figure 5: **R-MHE Validation: Isotropy and 2D Image Regression.** We analyze the impact of increasing the effective number of rotations $M$. **(a)** Isotropy vs M. The Anisotropy Ratio decreases and then increases as $M$ increases, demonstrating a more isotropic PSF for moderate $M$. Colors indicate the amplitude level at which the anisotropy ratio is measured (e.g., 0.5 corresponds to FWHM). **(b)** Visualization of the PSF zoom for different $M$. The shape becomes more circular (isotropic) as $M$ increases. **(c-e)** Qualitative comparison of 2D image regression results (zoomed view). R-MHE improves reconstruction quality by mitigating artifacts arising from the anisotropic kernel.

**Application: 2D Image Regression.** To demonstrate the practical advantages of R-MHE, we evaluated its performance on a 2D image regression task using three high-resolution images (Details in Appendix B). We follow the standard configuration: $L = 16$ levels and $F = 2$ features per level.

We employed our theoretical insight that the empirical FWHM (governed by $N_{\text{avg}}$ and $\beta_{\text{emp}}$) dictates the practical resolution limit (Section 3.3). Accordingly, we calculated the theoretical growth factor $b_{\text{theory}}$ such that the empirical FWHM along the axes direction, $\beta_{\text{emp}}/N_{\text{avg}}$, matches the spatial extent of a single pixel. We used the validated total empirical broadening factor $\beta_{\text{emp}} = 3.0$ (Section 3.2).

To validate this principled approach and find the empirical optimum, we performed experiments comparing $b_{\text{theory}}$ against neighboring values ($b_{\text{theory}} \pm 0.1, \pm 0.2$). The results (Appendix B.3, Table 5) show that the empirical optimum ($b_{\text{opt}}$) consistently occurred near, but slightly below, $b_{\text{theory}}$ (specifically $b_{\text{theory}} - 0.1$ or $-0.2$). This indicates the optimal effective kernel size is 2.5 pixels. This is physically intuitive, as natural images possess spatial coherence and are rarely composed of pixel-perfect high-frequency noise; a slightly broader kernel better matches the signal bandwidth and provides beneficial regularization.

We compare standard MHE ($M = 1$) and R-MHE ($M \in \{2, 4, 8\}$) using the respective empirically optimized $b_{\text{opt}}$ for each configuration. All experiments were run with 5 random seeds. The results, summarized in Table 1 and illustrated in Figure 5(c-e), highlight the improvement achieved by R-MHE. The standard MHE baseline achieves an average PSNR of 23.88 dB. R-MHE consistently improves performance, reaching a peak average PSNR of 24.82 dB at $M = 8$, an improvement of +0.94 dB. This gain demonstrates that the improved isotropy provided by R-MHE effectively mitigates artifacts caused by the anisotropic kernel, at no additional cost in parameters or computation.

Table 1: 2D Image Regression performance (Average PSNR $\pm$ Std Dev in dB) using the empirically optimized growth factor $b$. R-MHE significantly outperforms the standard MHE baseline ($M = 1$) with zero overhead. (L=16, F=2).

| Method (Effective Rotations M) | Average PSNR (dB) ↑ |
|---|---|
| Standard MHE (M=1) | $23.88 \pm 0.02$ |
| R-MHE (M=2) | $24.62 \pm 0.01$ |
| R-MHE (M=4) | $24.69 \pm 0.01$ |
| R-MHE (M=8) | $\mathbf{24.82 \pm 0.01}$ |

## 6 APPLICATIONS TO 3D NEURAL FIELDS AND SDFS

Having validated the benefits of R-MHE in 2D, we now evaluate its impact on complex 3D applications: Neural Radiance Fields (NeRF) and Signed Distance Functions (SDFs). We aim to demonstrate the practical utility of our PSF analysis for hyperparameter selection and to assess the performance of R-MHE in 3D using the polyhedral rotation strategy. We utilize a customized implementation based on the Instant-NGP framework (Müller et al., 2022).

### 6.1 EXPERIMENTAL SETUP

We follow the standard Instant-NGP configuration ($L = 16, F = 2$). All models were trained for 20,000 steps using the Adam optimizer (Details in Appendix C). We compare standard MHE against R-MHE using the 3D polyhedral rotation strategies (Tetrahedron, Cube, Octahedron, Icosahedron). All experiments were run with 5 random seeds. We evaluate two strategies for selecting the growth factor $b$. The Baseline Heuristic utilizes the default approach in Instant-NGP. The PSF Guided (Theory) strategy is derived from our analysis (Section 3); $b$ is calculated such that the empirical FWHM (using $\beta_{\text{emp}} = 3.0$) matches the target spatial resolution.

### 6.2 RESULTS: NEURAL RADIANCE FIELDS (NERF)

We conduct experiments on the 8 scenes of the Synthetic NeRF dataset (Mildenhall et al., 2020). The average reconstruction quality (PSNR) is summarized in Table 2.

Table 2: 3D NeRF reconstruction performance (Average PSNR $\pm$ Std Dev in dB) on the Synthetic NeRF dataset (8 scenes). We compare configuration strategies and R-MHE rotation types.

| Configuration | Method | Average PSNR (dB) ↑ |
|---|---|---|
| | Standard MHE | $35.346 \pm 0.105$ |
| | R-MHE (Tetra) | $35.472 \pm 0.114$ |
| Baseline Heuristic (Optimized $b$) | R-MHE (Cube) | $35.445 \pm 0.134$ |
| | R-MHE (Octa) | $35.449 \pm 0.115$ |
| | R-MHE (Icosa) | $\mathbf{35.479 \pm 0.134}$ |
| | Standard MHE | $35.329 \pm 0.100$ |
| | R-MHE (Tetra) | $35.396 \pm 0.128$ |
| PSF Guided (Theory $b$) | R-MHE (Cube) | $35.404 \pm 0.121$ |
| | R-MHE (Octa) | $35.409 \pm 0.139$ |
| | R-MHE (Icosa) | $\mathbf{35.440 \pm 0.119}$ |

The Baseline Heuristic (empirically optimized $b_{opt}$) performs very similarly to the PSF Guided (Theory) configuration ($b_{theory} \approx 1.38$). For Standard MHE, the theoretical configuration achieves 35.329 dB, matching the empirical optimum of 35.346 dB (Table 2). As visualized in the detailed parameter sweeps in Appendix C.2, the theoretical prediction ($b_{\text{theory}} \approx 1.38$) consistently falls precisely within the regime of best performance across all scenes. This validates that our PSF analysis, successfully identifies optimal hyperparameters a priori. R-MHE consistently maintains or slightly improves the performance of standard MHE across both configuration strategies. For the Baseline configuration, the best R-MHE variant (Icosa, 35.479 dB) shows a marginal improvement compared

to Standard MHE (35.346 dB). These improvements are often within one standard deviation of the baseline variance, and the quantitative performance gains in standard 3D benchmarks are statistically marginal compared to the 2D regression tasks. Nonetheless, R-MHE provides these results at zero overhead while offering the theoretical benefit of improved isotropy.

## 6.3 RESULTS: SIGNED DISTANCE FUNCTIONS (SDF)

We further evaluate R-MHE on the task of learning SDFs. We utilize three standard benchmark meshes: Armadillo, Bunny, and Spot. We measure the Intersection over Union (IoU) to quantify reconstruction quality (higher is better). The results for the optimal resolution configuration ($b \approx 1.18$) are summarized in Table 3. The results exhibit clear performance saturation. At the high spatial resolutions provided by the MHE configuration, the capacity of the hash grid is sufficient to resolve the geometry, yielding IoU scores exceeding $0.996$ for all meshes across all methods, and the reconstruction error is dominated by the finite sampling resolution rather than the encoding's anisotropy.

Table 3: 3D SDF reconstruction performance (Intersection over Union, IoU $\uparrow$) on benchmark meshes. All methods achieve near-perfect reconstruction, indicating performance saturation.

| Method | Armadillo | Bunny | Spot | Average IoU |
|---|---|---|---|---|
| Standard MHE | $0.9994 \pm 0.0002$ | $0.9966 \pm 0.0001$ | $0.9998 \pm 0.0001$ | 0.9986 |
| R-MHE (Tetra) | $0.9994 \pm 0.0002$ | $0.9966 \pm 0.0001$ | $0.9998 \pm 0.0001$ | 0.9986 |
| R-MHE (Cube) | $0.9995 \pm 0.0001$ | $0.9966 \pm 0.0001$ | $0.9998 \pm 0.0001$ | 0.9986 |
| R-MHE (Octa) | $0.9995 \pm 0.0001$ | $0.9966 \pm 0.0001$ | $0.9998 \pm 0.0001$ | 0.9986 |
| R-MHE (Icosa) | $0.9994 \pm 0.0002$ | $0.9966 \pm 0.0001$ | $0.9998 \pm 0.0001$ | 0.9986 |

## 7 CONCLUSION

This work addresses the deficiency of physical understanding in the study of Multi-Resolution Hash Encoding (MHE) by introducing an analysis based on the Point Spread Function (PSF). By treating the MHE model as a physical system, we provide a novel framework to identify performance limitations and optimize the architecture. Our analysis reveals that the idealized MHE PSF, characterized by the induced B-spline kernel, inherently possesses an anisotropic profile (broader on diagonals), with its spatial bandwidth determined by the average resolution $N_{\text{avg}}$. We demonstrated that optimization dynamics (spectral bias) lead to significant spatial broadening ($\beta_{\text{opt}}$), resulting in the crucial finding that the effective resolution is governed by the broadened empirical FWHM (and therefore $N_{\text{avg}}$), not $N_{\text{max}}$. Leveraging these insights, we introduced Rotated MHE (R-MHE), a parameter-free modification that mitigates anisotropy by applying distinct rotations at each resolution level.

**Limitations of R-MHE**    While R-MHE provides a theoretical guarantee of improved isotropy, we observe that the quantitative gains in standard 3D benchmarks are statistically marginal compared to the improvements seen in 2D tasks. We attribute this dampened effect to two primary factors inherent to these 3D tasks. First, in volumetric rendering (NeRF), the integration of features along rays acts as a form of "view averaging," which can effectively low-pass filter the high-frequency artifacts caused by grid anisotropy. Second, standard benchmarks often exhibit performance saturation at the high spatial resolutions typically employed, where the grid capacity is sufficient to resolve geometry regardless of orientation. Consequently, R-MHE is likely most critical in regimes with strict memory constraints or sparse viewing angles where interpolation artifacts are less likely to be masked by averaging or high sampling rates.

**Future Work and Broader Impact.**    The anisotropic behavior analyzed here is applicable to other grid-based encodings, such as multi-plane (TensoRF (Chen et al., 2022)) or planar factorizations (K-Planes (Fridovich-Keil et al., 2023)), as they also rely on multilinear interpolation on axis-aligned structures. The R-MHE concept of applying rotations per level or plane could be directly transferred to these settings to improve isotropy. This study establishes a physics-based methodology moving beyond heuristics to characterize and improve neural field models.

ACKNOWLEDGMENTS

This work was supported by the Samsung Global Outreach Program and the National Science Foundation under Award Number 2103301.

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

## A  EXTENDED DERIVATIONS

### A.1  IMPACT OF GRID MISALIGNMENT AND THE INDUCED KERNEL

The analysis of the idealized PSF (Section 3.1) relies on the induced kernel framework. In the linearized regime, the spatial response of the encoding averaged over all possible alignments between the target point $\mathbf{x}_0$ and the grid structure is characterized by the auto-correlation of the interpolation kernel $K_l(\mathbf{x})$, resulting in the induced kernel $B_l(\mathbf{x}) = (K_l * K_l)(\mathbf{x})$ (the cubic B-spline). While the response for a specific alignment (e.g., perfect alignment with a vertex, which yields the tent function) differs, the B-spline kernel correctly captures the general case, and the derived properties regarding anisotropy and FWHM scaling are robust.

### A.2  CONTEXT: BEHAVIOR UNDER DENSE SUPERVISION

It is important to contrast the behavior of MHE under sparse supervision, which is the focus of our PSF analysis, with its behavior under dense supervision, such as in image regression tasks. Our analysis, however, specifically targets the behavior relevant to scenarios with sparse constraints, where the assumption of a linearized MLP decoder is most appropriate and the intrinsic spatial properties of the encoding are isolated, and by our analysis, closely links the final performance on actual tasks with this idealized results.

### A.3  ANALYSIS OF THE INDUCED KERNEL: PROFILE, ANISOTROPY, AND FWHM

We analyze the properties of the idealized PSF $P_{\text{Ideal}}(\mathbf{x})$ based on the induced kernel $B(\mathbf{x})$ (the separable cubic B-spline kernel) and the multi-resolution structure.

**The 1D Cubic B-spline Kernel and FWHM ($\beta_{\text{ideal}}$).** In 1D, the induced kernel is the normalized cubic B-spline $\hat{B}_{1D}(u)$. The piecewise definition for $0 \le |u| \le 1$ is:

$$\hat{B}_{1D}(u) = 1 - \frac{3}{2}u^2 + \frac{3}{4}|u|^3 \tag{8}$$

We solve for the half-width $u_{1/2}$ such that $\hat{B}_{1D}(u_{1/2}) = 0.5$. Solving this cubic equation numerically yields the relevant real root $u_{1/2} \approx 0.5904$. The FWHM is $\beta_{\text{ideal}} = 2u_{1/2} \approx 1.1808$.

**Derivation of the Logarithmic Profile and Closed Form.** We demonstrate that the superposition of B-splines results in a dominant logarithmic profile near the center, justifying Eq. 2. We approximate the summation with an integral and utilize the Taylor expansion of the B-spline near the center: $\hat{B}_{1D}(u) \approx 1 - \frac{3}{2}u^2$.

We analyze the 1D case. Let $v = N_{\min}x$. The effective upper limit of integration $L_{eff}$ occurs approximately when the argument reaches the scale of the kernel support, $b^{L_{eff}}v \approx 1$. Thus $L_{eff} \approx -\ln(v)/\ln(b)$.

$$P_{\text{Ideal}}(v) \approx \frac{1}{L}\int_0^{L_{eff}} \hat{B}_{1D}(b^l v)dl \approx \frac{1}{L}\int_0^{L_{eff}}\left(1 - \frac{3}{2}b^{2l}v^2\right)dl \tag{9}$$

$$P_{\text{Ideal}}(v) \approx \frac{1}{L}\left[l - \frac{3v^2}{4\ln b}b^{2l}\right]_0^{L_{eff}} \tag{10}$$

Substituting $L_{eff}$ and $b^{2L_{eff}} \approx 1/v^2$:

$$P_{\text{Ideal}}(v) \approx \frac{1}{L\ln b}\left[-\ln(v) - \frac{3}{4} + \frac{3v^2}{4}\right] \tag{11}$$

This confirms the dominant $-\ln(v)$ behavior near the center. The generalization to D dimensions yields the form in Eq. 2, where the anisotropy factor $A_D(\mathbf{v})$ arises from the D-dimensional expansion and the specific properties of the B-spline kernel.

**FWHM Scaling.** The FWHM of the composite idealized PSF $P_{\text{Ideal}}(\mathbf{x})$ along the axes scales with the average resolution $N_{\text{avg}}$. The idealized FWHM along the axis is approximately:

$$\text{FWHM}_{\text{Axis, Ideal}} \approx \beta_{\text{ideal}}/N_{\text{avg}} \approx 1.18/N_{\text{avg}} \tag{12}$$

### A.4 GENERALIZED ANISOTROPY ANALYSIS IN D DIMENSIONS

We perform the anisotropy analysis of the induced kernel (cubic B-spline) on $D$ dimensions using an analysis based on the Taylor expansion near the center. This analysis demonstrates that the kernel is inherently narrowest along the primary axes and becomes progressively broader along directions that involve multiple components (diagonals).

**D-Dimensional Kernel and Taylor Approximation.** The D-dimensional normalized induced kernel $\hat{B}_D(\mathbf{u})$ is the separable product:

$$\hat{B}_D(\mathbf{u}) = \prod_{i=1}^{D} \hat{B}_{1D}(u_i) \tag{13}$$

We utilize the Taylor expansion of the 1D kernel near the origin. From the definition of the normalized cubic B-spline (Eq. 8), the second derivative is $\hat{B}''_{1D}(0) = -3$. We define $C = -\frac{1}{2}\hat{B}''_{1D}(0) = 3/2$.

$$\hat{B}_{1D}(u) \approx 1 - Cu^2 \tag{14}$$

This quadratic approximation accurately captures the local curvature near the peak, which determines the primary anisotropy trend.

**Analysis of K-Sparse Directions.** We analyze the anisotropy by comparing the Euclidean distance $d$ from the center required to reach a fixed amplitude $A$ (e.g., $A = 0.5$ for FWHM) along different orientations. We consider a $K$-sparse direction, where $K$ components of the input vector are equal ($u_i = x$ for $1 \le i \le K$) and the remaining $D - K$ components are zero. This spans orientations from the primary axis ($K = 1$) to the main space diagonal (if $K = D$). The Euclidean distance is $d = \sqrt{K}x$.

The response along this direction is:

$$P(\mathbf{u}) = \prod_{i=1}^{K}\hat{B}_{1D}(x) \cdot \prod_{i=K+1}^{D}\hat{B}_{1D}(0) = \left(\hat{B}_{1D}(x)\right)^K \tag{15}$$

Setting the response to $A$:

$$\hat{B}_{1D}(x)^K = A \implies \hat{B}_{1D}(x) = A^{1/K} \tag{16}$$

Using the Taylor expansion:

$$1 - Cx^2 \approx A^{1/K} \implies x^2 \approx \frac{1 - A^{1/K}}{C} \tag{17}$$

The squared Euclidean distance $d^2$ is:

$$d^2(K) = Kx^2 \approx \frac{K(1 - A^{1/K})}{C} \tag{18}$$

**Proof of Monotonicity.** To determine how the width changes as the direction involves more components (moving from axis towards the main diagonal), we analyze the monotonicity of $d^2(K)$. We analyze the function $f(K) = K(1 - A^{1/K})$ for $A \in (0, 1)$.

We analyze the derivative $f'(K)$ (treating K as a continuous variable for analysis):

$$f'(K) = (1 - A^{1/K}) + K\left(-A^{1/K} \cdot \ln(A) \cdot \left(-\frac{1}{K^2}\right)\right) = 1 - A^{1/K} + \frac{1}{K}A^{1/K}\ln(A) \tag{19}$$

Let $y = A^{1/K}$. Since $A \in (0, 1)$ and $K \geq 1$, we have $y \in (0, 1)$. Also $\ln(A) = K\ln(y)$.

$$f'(K) = 1 - y + y\ln(y) \tag{20}$$

We examine the function $g(y) = 1 - y + y\ln(y)$. The derivative is $g'(y) = -1 + \ln(y) + 1 = \ln(y)$. Since $y < 1$, $g'(y) < 0$. Thus, $g(y)$ is monotonically decreasing. The minimum occurs at the limit $y \to 1$, where $g(1) = 0$. Therefore, $f'(K) > 0$ for all $A < 1$.

This proves that $f(K)$, and thus the distance $d(K)$, is monotonically increasing with $K$. The induced B-spline kernel is narrowest along the primary axes ($K = 1$) and broadest along the main space diagonal (where $K$ is maximal).

**Anisotropy Ratio and Dimensional Scaling.** We define the Anisotropy Ratio $R_A(D)$ as the ratio of the squared distances along the main diagonal ($K = D$) versus the axis ($K = 1$), using $A = 0.5$.

$$R_A(D) = \frac{d^2(D)}{d^2(1)} \approx \frac{D(1 - 0.5^{1/D})/C}{(1 - 0.5)/C} = 2D(1 - 0.5^{1/D}) \tag{21}$$

- **D=2:** $R_A(2) = 4(1 - 0.5^{1/2}) \approx 1.1716$.
- **D=3:** $R_A(3) = 6(1 - 0.5^{1/3}) \approx 1.2378$.

The Anisotropy Ratio $R_A(D)$ is also monotonically increasing with $D$ (since $f(D)$ is monotonic), demonstrating that the grid-induced anisotropy becomes more pronounced in higher dimensions. The limit as $D \to \infty$ is $2\ln(2) \approx 1.386$.

## A.5 TWO-POINT RESOLUTION DERIVATION (CONSTRUCTIVE INTERFERENCE)

The resolution limit $d_{\text{crit}}$ (Rayleigh criterion) is the minimum separation $d$ such that $R(\text{midpoint}) < R(\text{peak})$. We analyze the idealized case for an axis-aligned configuration. The condition for a dip at the midpoint requires the second derivative of the reconstruction $R(\mathbf{x})$ to be positive at the center. Since the induced kernel $B_l(\mathbf{x})$ (cubic B-spline) is smooth ($C^2$ continuous) and strictly concave near the peak, this condition is satisfied for any separation $d > 0$.

Therefore, in the idealized minimum-norm configuration using the induced kernel, the theoretical resolution limit is infinitesimally small. It is crucial to emphasize that this applies specifically to the idealized solution. As demonstrated by the empirical results in Section 3.3, practical optimization dynamics lead to a broader empirical PSF, where the actual resolution limit is governed by the empirical FWHM (scaling with $1/N_{\text{avg}}$).

A.6  R-MHE 3D ROTATION STRATEGIES

We detail the 3D rotation strategies utilized in Section 5.2. We use the vertex orientations of regular polyhedra to define canonical directions, aiming for uniform sampling of SO(3). The rotations $\mathbf{R}_l$ are constructed by aligning the standard basis vectors with these normalized directions. We cycle through the following sets across the levels $L$.

- **Tetrahedron (4 vertices):** (1,1,1), (-1,-1,1), (-1,1,-1), (1,-1,-1).
- **Cube (8 vertices):** All sign corners $(\pm1, \pm1, \pm1)$, ordered lexicographically.
- **Octahedron (6 vertices):** $(\pm1, 0, 0), (0, \pm1, 0), (0, 0, \pm1)$.
- **Icosahedron (12 vertices):** $(0, \pm1, \pm\phi), (\pm1, \pm\phi, 0), (\pm\phi, 0, \pm1)$, where $\phi = (1+\sqrt{5})/2$ is the golden ratio.

The Icosahedral strategy provides the most uniform distribution of orientations, which correlates with the superior empirical performance observed in Section 6.

## B  EXPERIMENTAL DETAILS FOR 2D IMAGE REGRESSION

### B.1  DATASET AND CONFIGURATION

We utilized three high-resolution images: "Mountain" ($2473 \times 3710$), "City" ($4000 \times 6000$), and "Forest" ($3193 \times 6016$). To ensure fair comparison and remove potential anisotropy introduced by non-square aspect ratios during sampling, all images were center-cropped to the largest possible square (e.g., $2473 \times 2473$ for Mountain).

**Configuration.**  We utilized $L = 16$ levels, $F = 2$ features per level, and a hash table capacity of $T = 2^{18}$. R-MHE configurations use the progressive rotation strategy (Section 5.2), with $M$ defining the base rotation angle $\theta = 90°/M$.

**Training Details.**  The networks were trained for 5000 steps using the Adam optimizer with a learning rate of 0.001. We used a batch size of $2^{17}$ (131072) randomly sampled pixels per iteration. All experiments were repeated across 5 random seeds.

### B.2  PSF GUIDED PARAMETER SELECTION

We leveraged the relationship between the empirical FWHM along the axes direction and $N_{\text{avg}}$. We determined the theoretical optimal growth factor $b_{\text{theory}}$ by setting the target empirical FWHM ($\beta_{\text{emp}}/N_{\text{avg}}$) based on the cropped image dimensions (1/shortest_side). We used the empirically validated factor $\beta_{\text{emp}} = 3.0$. We fixed $L = 16$ and $N_{\text{min}} = 16$, and then solved numerically for the required growth factor $b$. The resulting theoretical growth factors $b_{\text{theory}}$ are detailed in Table 4.

Table 4: PSF Guided Hyperparameters for 2D Gigapixel Image Regression (Cropped Images).

| Image | Resolution (Cropped) | Target Empirical FWHM | Growth Factor ($b_{\text{theory}}$) |
|---|---|---|---|
| Mountain | $2473 \times 2473$ | 8.09e-04 | 1.4614 |
| City | $4000 \times 4000$ | 5.00e-04 | 1.5078 |
| Forest | $3193 \times 3193$ | 6.26e-04 | 1.4863 |

### B.3  VALIDATION OF THEORETICAL GROWTH FACTOR

The results (Table 5) summarize these results across the Standard MHE (M=1) and R-MHE configurations ($M \in \{2, 4, 8\}$). The analysis demonstrates that the best performance is consistently achieved near $b_{\text{theory}}$, specifically at $b_{\text{theory}} - 0.1$ or $-0.2$. Since a lower growth factor $b$ corresponds to a lower $N_{\text{avg}}$ and thus a wider FWHM, this finding suggests that targeting a resolution of exactly one pixel is slightly too aggressive. A kernel slightly broader than one pixel yields better reconstruction, likely because natural images are band-limited and contain features larger than a single pixel,

whereas a 1-pixel target may encourage overfitting to aliasing or quantization artifacts. The optimal FWHM is thus roughly 2.5 pixels, from this observation.

Table 5: Validation of PSF-Guided Growth Factor ($b$) Selection. PSNR (dB, Mean $\pm$ Std Dev) results comparing $b_{\text{theory}}$ against neighboring values. Bold indicates the best performance for the configuration.

| Image | M | $b_{\text{theory}} - 0.2$ | $b_{\text{theory}} - 0.1$ | $b_{\text{theory}}$ | $b_{\text{theory}} + 0.1$ | $b_{\text{theory}} + 0.2$ |
|---|---|---|---|---|---|---|
| Mountain | M=1 | $22.98 \pm 0.02$ | $\mathbf{24.19 \pm 0.03}$ | $23.07 \pm 0.04$ | $22.44 \pm 0.09$ | $21.76 \pm 0.07$ |
| | M=2 | $23.90 \pm 0.01$ | $\mathbf{25.37 \pm 0.02}$ | $24.52 \pm 0.01$ | $23.30 \pm 0.02$ | $22.75 \pm 0.01$ |
| | M=4 | $24.09 \pm 0.01$ | $\mathbf{25.35 \pm 0.02}$ | $24.53 \pm 0.01$ | $23.59 \pm 0.01$ | $23.54 \pm 0.02$ |
| | M=8 | $24.12 \pm 0.00$ | $\mathbf{25.84 \pm 0.01}$ | $24.19 \pm 0.02$ | $23.40 \pm 0.02$ | $23.35 \pm 0.02$ |
| City | M=1 | $21.85 \pm 0.00$ | $\mathbf{21.86 \pm 0.01}$ | $21.16 \pm 0.01$ | $20.20 \pm 0.00$ | $19.07 \pm 0.00$ |
| | M=2 | $22.31 \pm 0.00$ | $\mathbf{22.57 \pm 0.01}$ | $21.71 \pm 0.01$ | $20.64 \pm 0.00$ | $19.77 \pm 0.00$ |
| | M=4 | $22.33 \pm 0.00$ | $\mathbf{22.71 \pm 0.00}$ | $21.66 \pm 0.01$ | $21.04 \pm 0.00$ | $20.44 \pm 0.01$ |
| | M=8 | $\mathbf{22.82 \pm 0.00}$ | $22.43 \pm 0.00$ | $21.62 \pm 0.01$ | $21.06 \pm 0.01$ | $20.46 \pm 0.01$ |
| Forest | M=1 | $25.11 \pm 0.01$ | $\mathbf{25.59 \pm 0.01}$ | $24.76 \pm 0.02$ | $24.46 \pm 0.03$ | $24.12 \pm 0.01$ |
| | M=2 | $25.56 \pm 0.01$ | $\mathbf{25.92 \pm 0.01}$ | $25.39 \pm 0.01$ | $25.19 \pm 0.01$ | $24.66 \pm 0.01$ |
| | M=4 | $25.55 \pm 0.01$ | $\mathbf{26.02 \pm 0.00}$ | $25.38 \pm 0.01$ | $25.13 \pm 0.01$ | $24.85 \pm 0.01$ |
| | M=8 | $\mathbf{25.79 \pm 0.01}$ | $25.78 \pm 0.00$ | $25.30 \pm 0.01$ | $25.04 \pm 0.01$ | $25.02 \pm 0.01$ |

### B.4 DETAILED RESULTS

Table 6 provides the detailed PSNR results for each individual image across the different R-MHE configurations, utilizing the empirically validated optimal $b$ (from Table 5).

Table 6: Detailed PSNR (dB, Mean $\pm$ Std Dev) results for 2D Gigapixel Image Regression using the empirically optimized growth factor $b$. Bold indicates the best performance per row.

| Image | M=1 (Baseline) | M=2 | M=4 | M=8 |
|---|---|---|---|---|
| Mountain | $24.19 \pm 0.03$ | $25.37 \pm 0.02$ | $25.35 \pm 0.02$ | $\mathbf{25.84 \pm 0.01}$ |
| City | $21.86 \pm 0.01$ | $22.57 \pm 0.01$ | $22.71 \pm 0.00$ | $\mathbf{22.82 \pm 0.00}$ |
| Forest | $25.59 \pm 0.01$ | $25.92 \pm 0.01$ | $\mathbf{26.02 \pm 0.00}$ | $25.79 \pm 0.01$ |
| Average | $23.88 \pm 0.02$ | $24.62 \pm 0.01$ | $24.69 \pm 0.01$ | $\mathbf{24.82 \pm 0.01}$ |

### B.5 QUALITATIVE VISUALIZATION AND PSF ANALYSIS

**Qualitative Comparison.** Figure 6 provides a qualitative comparison of the reconstruction results.

**PSF Visualization.** Figure 7 provides visualizations of the empirical PSF for R-MHE with varying $M$. As $M$ increases, the PSF profile becomes noticeably more symmetric and circular. For larger $M$, a faint spiral pattern emerges, characteristic of the progressive rotation strategy employed in 2D R-MHE.

## C EXPERIMENTAL DETAILS FOR 3D NEURAL FIELDS AND SDFS

### C.1 CONFIGURATION AND TRAINING

We utilized the standard Instant-NGP configuration (L=16, F=2, $T = 2^{19}$). Training utilized the standard Instant-NGP protocol and optimizer settings. All experiments were run with 5 random seeds.

### C.2 NEURAL RADIANCE FIELDS (NERF) DETAILS

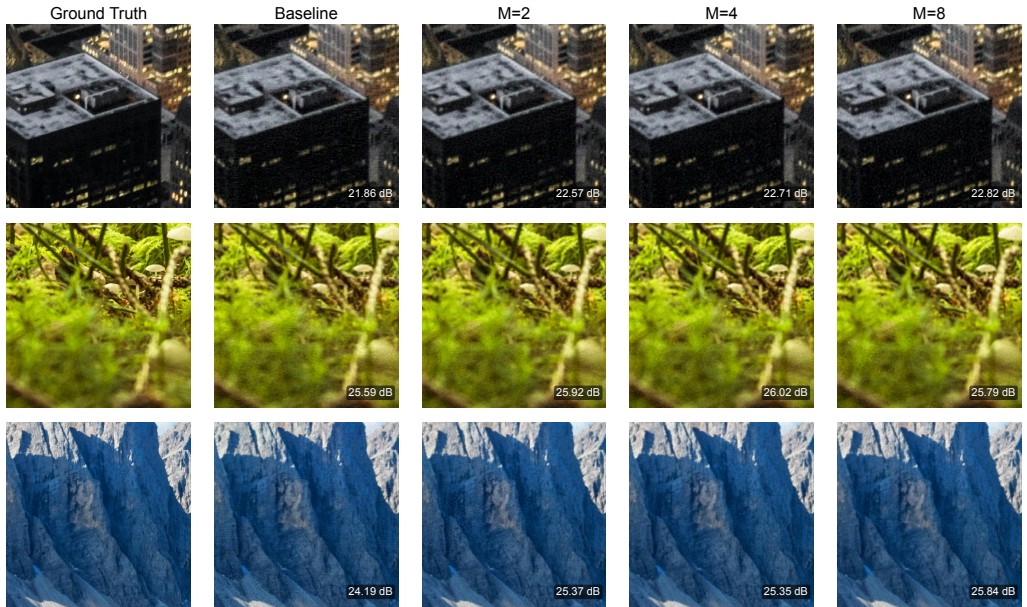

Figure 6: **Qualitative Visualization of 2D Image Regression Results.** Zoomed-in crops comparing Ground Truth, Standard MHE (Baseline), and R-MHE (M=2, 4, 8). R-MHE demonstrates significant improvements in reconstruction quality. PSNR values indicated are for the specific runs shown.

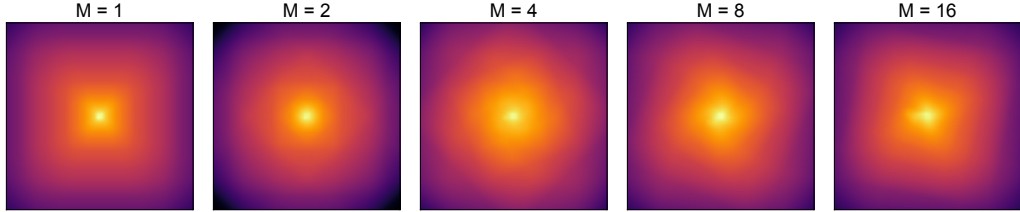

Figure 7: **Visualization of R-MHE Point Spread Function (PSF).** Empirical PSF (zoomed view) for R-MHE with M=1 (Standard MHE), 2, 4, 8, and 16.

We utilized all eight scenes from the Synthetic NeRF dataset. Figure 8 presents a comprehensive parameter sweep of the reconstruction quality (PSNR) as a function of the growth factor $b$ for all eight NeRF scenes.

These plots demonstrate that the theoretical growth factors derived from our PSF analysis ($b_{\text{theory}} \approx 1.38$, as listed in Table 7) consistently fall within the regime of best performance. As observed in the figure, the empirical peaks for scenes such as *Lego*, *Mic*, and *Hotdog* align precisely with the theoretical prediction of 1.38. This confirms that our physical systems approach accurately characterizes the optimal spatial resolution settings, allowing for precise hyperparameter selection without the need for exhaustive empirical grid searches.

## C.3    SIGNED DISTANCE FUNCTIONS (SDF) DETAILS

**Dataset.**    We utilized three standard benchmark meshes: Armadillo, Bunny, Spot. We used the baseline parameters provided in the Instant-NGP repository for SDF tasks.

Figure 9 visualizes the IoU performance across the growth factor sweep for the SDF task.

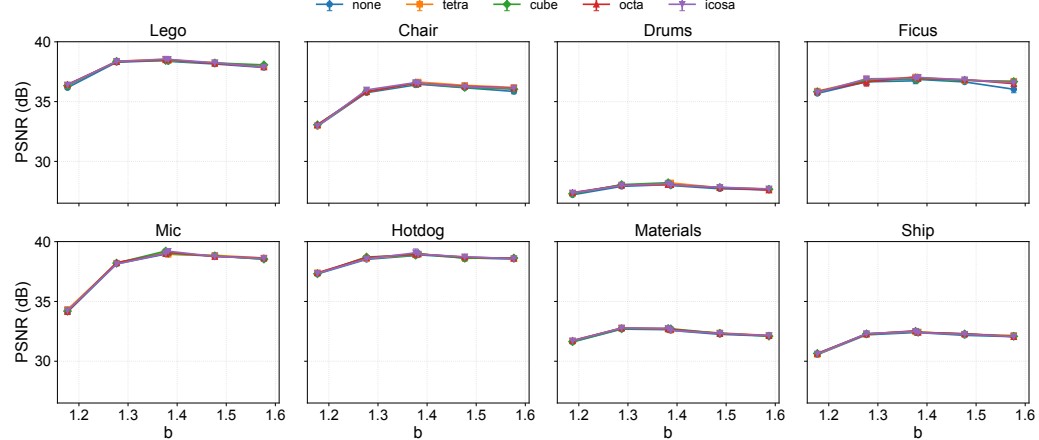

Figure 8: **Detailed PSNR Sweep for Synthetic NeRF Experiments.** We plot the Average PSNR (dB) vs. growth factor $b$ across all 8 scenes (Error bars indicate Std Dev over 5 seeds). The theoretical optimum derived in our analysis ($b_{\text{theory}} \approx 1.38$) consistently aligns with the empirical peak performance observed in these sweeps, validating the PSF-guided strategy. The R-MHE strategies (tetra, cube, octa, icosa) consistently track or exceed the performance of the Standard MHE baseline ('none').

Table 7: Growth Factors (b) for Synthetic NeRF Experiments. $b_{\text{theory}}$ is the value derived from the analysis framework. $b_{\text{opt}}$ is the empirically optimized value used in the Baseline Heuristic configuration.

| Scene | $b_{\text{theory}}$ | $b_{\text{opt}}$ **(Baseline)** |
|---|---|---|
| Chair | 1.3767 | 1.3819 |
| Drums | 1.3872 | 1.3819 |
| Ficus | 1.3767 | 1.3819 |
| Hotdog | 1.3767 | 1.3819 |
| Lego | 1.3767 | 1.3819 |
| Materials | 1.3872 | 1.3819 |
| Mic | 1.3767 | 1.3819 |
| Ship | 1.3767 | 1.3819 |

# D THEORETICAL DERIVATION AND EMPIRICAL ANALYSIS OF THE BROADENING FACTOR

## D.1 MODELING OPTIMIZATION DYNAMICS VIA SPECTRAL BIAS

The idealized Point Spread Function (PSF), derived under the minimum L2-norm assumption (Eq. 1), implies uniform weighting across all $L$ resolution levels:

$$P_{\text{Ideal}}(\mathbf{x}) = \frac{1}{L} \sum_{l=0}^{L-1} \hat{B}_l(\mathbf{x}) \tag{22}$$

This leads to the idealized FWHM factor $\beta_{\text{ideal}} \approx 1.18$.

Empirical results demonstrate that gradient-based optimization methods (e.g., Adam) exhibit spectral bias, where lower frequencies are learned preferentially (Rahaman et al., 2018). In the context of MHE, this biases the optimization towards coarser grids (low $N_l$). We model the resulting empirical PSF as a weighted superposition with non-uniform weights $w_l$:

$$P_{\text{Empirical}}(\mathbf{x}) = \sum_{l=0}^{L-1} w_l \hat{B}_l(\mathbf{x}) \tag{23}$$

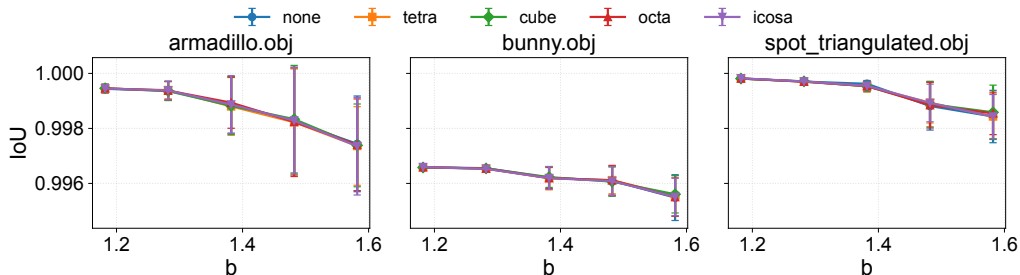

Figure 9: **Detailed IoU Sweep for 3D SDF Experiments.** We plot the Intersection over Union (IoU) as a function of the growth factor $b$ for the Armadillo, Bunny, and Spot meshes. The heavy overlap of the curves for all methods (Standard MHE and R-MHE variants) and the consistently high IoU values ($> 0.996$) illustrate the performance saturation discussed in the main text.

where $\sum w_l = 1$. This leads to the total empirical FWHM factor $\beta_{\text{emp}}$.

We adopt a phenomenological model motivated by kernel methods, where the optimization-induced weights follow a power law relationship based on the resolution $N_l$:

$$w_l \propto (N_l)^{-\gamma} \tag{24}$$

Here, $\gamma \geq 0$ is the spectral bias exponent. $\gamma = 0$ recovers the minimum-norm solution ($\beta_{\text{ideal}}$), while $\gamma > 0$ indicates a bias towards coarser levels ($\beta_{\text{emp}} > \beta_{\text{ideal}}$).

### D.2 RELATIONSHIP BETWEEN SPECTRAL BIAS AND BROADENING

We establish that the optimization-induced broadening factor $\beta_{\text{opt}}(\gamma)$ (where $\beta_{\text{emp}} = \beta_{\text{ideal}} \cdot \beta_{\text{opt}}$) is a monotonically increasing function of the spectral bias exponent $\gamma$.

**Proof of Monotonicity of $\beta_{\text{opt}}(\gamma)$.** We want to show that $d(\text{FWHM})/d\gamma > 0$. We analyze the empirical PSF value at a fixed spatial position $x > 0$, $P(\gamma) = P_{\text{Empirical}}(x; \gamma)$. If we show that $dP(\gamma)/d\gamma > 0$ (i.e., the tails of the PSF increase with $\gamma$), it implies the FWHM must also increase.

We express $P(\gamma)$ using normalized weights $w_l = (N_l)^{-\gamma} / \sum_k (N_k)^{-\gamma}$.

$$P(\gamma) = \frac{\sum_l (N_l)^{-\gamma} \hat{B}_l(x)}{\sum_k (N_k)^{-\gamma}} = \frac{f(\gamma)}{g(\gamma)} \tag{25}$$

We analyze the derivative $P'(\gamma) = (f'g - fg')/g^2$. We must show $f'g - fg' > 0$.

$$f'(\gamma) = -\sum_l (N_l)^{-\gamma} \ln(N_l) \hat{B}_l(x) \tag{26}$$

$$g'(\gamma) = -\sum_k (N_k)^{-\gamma} \ln(N_k) \tag{27}$$

We analyze the term $f'g - fg'$:

$$f'g - fg' = \left( -\sum_l (N_l)^{-\gamma} \ln N_l \hat{B}_l(x) \right) \left( \sum_k (N_k)^{-\gamma} \right) - \left( \sum_l (N_l)^{-\gamma} \hat{B}_l(x) \right) \left( -\sum_k (N_k)^{-\gamma} \ln N_k \right) \tag{28}$$

$$= \sum_{l,k} (N_l)^{-\gamma} (N_k)^{-\gamma} \hat{B}_l(x) (\ln N_k - \ln N_l) \tag{29}$$

We analyze this summation by grouping pairs of indices $(l, k)$ and $(k, l)$ where $l \neq k$:

$$\text{Pair}_{(l,k)} = (N_l)^{-\gamma}(N_k)^{-\gamma}\hat{B}_l(x)(\ln N_k - \ln N_l) + (N_k)^{-\gamma}(N_l)^{-\gamma}\hat{B}_k(x)(\ln N_l - \ln N_k) \tag{30}$$

$$= (N_l)^{-\gamma}(N_k)^{-\gamma}(\ln N_k - \ln N_l)\left[ \hat{B}_l(x) - \hat{B}_k(x) \right] \tag{31}$$

Consider the case $N_k > N_l$. The factor $(\ln N_k - \ln N_l)$ is positive. Since $N_k > N_l$ (level $k$ is finer), the kernel $\hat{B}_k(x)$ is narrower than $\hat{B}_l(x)$. For $x > 0$, we have $\hat{B}_l(x) > \hat{B}_k(x)$. Thus, the second factor is also positive.

Every pair sum is positive, proving that $f'g - fg' > 0$. Consequently, $dP(\gamma)/d\gamma > 0$, which implies $d\beta_{\mathrm{opt}}/d\gamma > 0$. A stronger spectral bias always leads to larger broadening.

### D.3 EMPIRICAL ANALYSIS OF OPTIMIZER DEPENDENCE

We investigated the dependency of the broadening factor on the choice of optimizer. We trained the MHE network using various standard optimizers (Adam variants, SGD, RMSProp, etc.) and measured the resulting total empirical broadening $\beta_{\mathrm{emp}}$. Figure 10 summarizes the results.

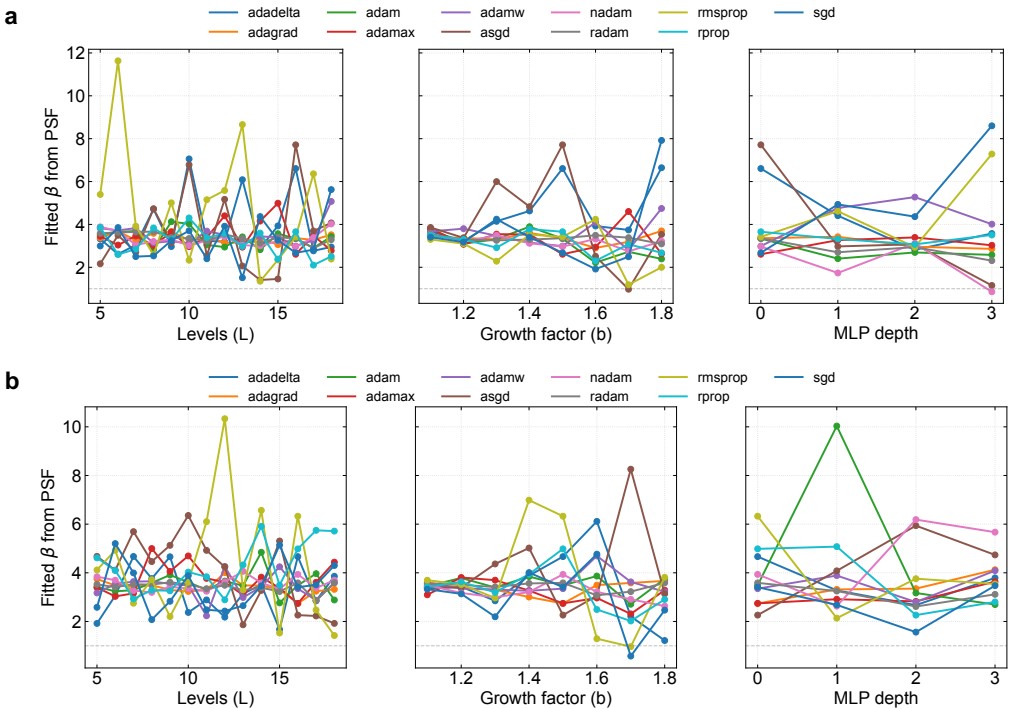

Figure 10: **Sensitivity of Broadening Factor to Optimizer Choice.** We measure the fitted total empirical broadening $\beta_{\mathrm{emp}}$ across different optimizers and MHE configurations (varying $L$ and $b$). **(a)** Results for 2D. **(b)** Results for 3D. Broadening depends significantly on the optimizer (Adam variants consistently yield $\beta_{\mathrm{emp}} \approx 3.0$), but is robust across different MHE parameters and dimensions for a given optimizer.

We observe that the broadening is indeed dependent on the optimizer. Adam and its variants (AdamW, NAdam, Adamax) consistently yield $\beta_{\mathrm{emp}} \approx 3.0$ (corresponding to $\beta_{\mathrm{opt}} \approx 2.54$), with 3D results almost identical to 2D. Other optimizers exhibit different degrees of broadening, as different optimization algorithms inherently possess different implicit regularization properties. For a given optimizer, the resulting $\beta_{\mathrm{emp}}$ is highly robust across different MHE configurations (varying $L$ and $b$) and dimensions (2D vs 3D), as shown in Figure 10. Panels (a) and (b) show results for 2D and 3D respectively, both yielding similar broadening values for each optimizer, confirming that the broadening is primarily determined by the optimizer choice, rather than specific MHE parameters. This stability means that a one-time calibration of $\beta_{\mathrm{emp}}$ for a specific optimization setup is sufficient for principled hyperparameter selection.

### D.4 SENSITIVITY TO MLP ARCHITECTURE

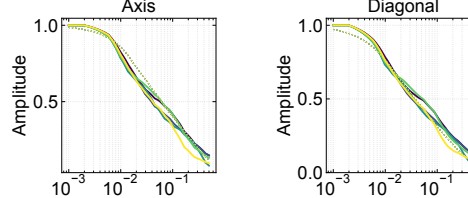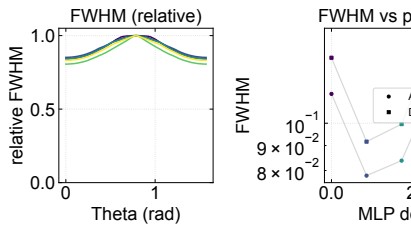

Figure 11: **Impact of MLP Depth on the Empirical PSF (2D).** We analyze the PSF while varying the MLP depth (indicated by color). The results show that the empirical PSF profile and the FWHM are largely insensitive to the MLP architecture.

We investigated the impact of the MLP decoder depth on the empirical PSF. Figure 11 shows the results of varying the MLP depth from 0 (linear) to 3 layers. The empirical PSF profiles and the resulting FWHM are consistent across different depths, supporting the conclusion that the encoding structure and the optimization dynamics (spectral bias) primarily define the spatial response.

