# OpenReview forum: "Characterizing and Optimizing the Spatial Kernel of Multi Resolution Hash Encodings"
_ICLR.cc/2026/Conference — ICLR 2026 Poster_

### Official Review · Reviewer_QjpZ · 2025-10-30

**Soundness:** 3
**Presentation:** 3
**Contribution:** 3
**Rating:** 4
**Confidence:** 2

**Summary:**

This paper introduces a method to analyze multi-resolution hash encoding used in implicit neural representations. The authors observe that while MHE is widely adopted, its hyperparameters have been tuned empirically with little mathematical analysis of its behavior. Thus, this paper introduce a point spread function based theoreritcal framework to analyze how MHE operates. Under multi-resolution, they prove that the influence region of each point feature scales with 𝑁min; they also study differences that arise during optimization and propose a model to explain them. They further propose a speckle/noise approximation to model hash collisions, and introduce R-MHE to mitigate these issues. On high-resolution image regression and 3D NeRF training, they show that the analysis is consistent with practice and yields useful hyperparameter guidance.

**Strengths:**

Motivation: While practitioners recognize certain behaviors of hash encodings, providing a mathematical model and using it to select hyperparameters is a meaningful contribution.

Clarity of modeling: The paper steps through the math cleanly, organizing the required formulas at each stage.

**Weaknesses:**

**3D experiment observations**
- Even when values are similar, the baseline consistently outperforming the theory in 3D suggests some assumptions may not hold in 3D. Could this stem from the alignment assumption in the theory? A toy misalignment study (small perturbations) could demonstrate how trends evolve and help validate the proposed modeling more precisely.

**Theoretical growth factor experiments.**
- Results are shown by varying b with ±0.1 steps, but: NeRF training is stochastic; single runs are not sufficient—use N repeated trials to report robustness. Use wider resolution than 0.1 steps to confirm the monotonic/expected trend across denser b values.

**Practical calibration of the broadening factor.**
- Does the broadening factor depend on training iterations or optimizer choice? If training dynamics materially affect it—even if grids don’t—then practical calibration may be bottleneck for real applying this analysis on real hyperparameter tuning.

**Questions:**

- The anisotropic behavior seems applicable to other encodings (e.g., multi-plane or multi-resolution plane encodings). Do the authors have a unified formulation or insights that transfer to those settings?

---

> ### Author Response · Authors · 2025-11-23
>
> We thank the reviewer for acknowledging the clarity of our modeling and the contribution of our analysis.
>
> W1: 3D experiment observations (Baseline vs Theory).
>
> This "alignment assumption" is indeed important for consideration. In the initial submission, the idealized PSF analysis relied on an assumption of perfect alignment. We have addressed this limitation in the revision by adopting the induced kernel (B-spline) framework (Section 3.1, Appendix A.1). This framework rigorously characterizes the spatial response averaged over all possible grid alignments. The revised theoretical analysis, including the new rigorous derivation of anisotropy (Appendix A.4), is based on this foundation and is robust to misalignment.
>
> In the 3D setup, the baseline heuristic performed similarly to the theory-derived. This occurred because the default heuristic happened to be extremely close to the theoretical value. The theory validates the heuristic in this specific case. The value of our theory is providing a principled way to derive near-optimal parameters in new scenarios without extensive grid search, as demonstrated in the 2D task. We also provide justification of the optimal values in the appendix by leveraging multiple runs with different b values.
> W2: Theoretical growth factor experiments (Robustness). Single runs are not sufficient.
>
> We have addressed this by rerunning all experiments (2D Image Regression, 3D NeRF, and the new 3D SDF task) across 5 random seeds. We have updated all tables (Tables 1-3 and Appendix tables) to report the mean and standard deviation. We also expanded the growth factor validation (Appendix B.3) to include $\pm 0.1$ and $\pm 0.2$ steps. The updated results confirm the trends reported previously, reinforcing the statistical significance of the findings.
>
> W3: Practical calibration of the broadening factor (bottleneck?).
>
> This is an important consideration. To investigate the dependency of $\beta$ on training dynamics, we conducted new experiments (Section 3.2 and Appendix D.4). We found $\beta$ is indeed dependent on the optimizer (e.g., $\beta \approx 3.0$ for Adam variants), consistent with its theoretical origin in spectral bias (Appendix D.1-D.3).
>
> We found that for a given optimizer, $\beta$ is stable across training iterations and highly robust across different MHE configurations ($L, b$). This stability means that the PSF-guided hyperparameter selection is highly practical and does not require re-calibration for different tasks. This means that an one-time calibration for a given optimizer configuration (e.g., $\beta=3$ for Adam) is sufficient, and it is not a practical bottleneck.
>
> Q1: Applicability to other encodings (e.g., multi-plane).
>
> Yes. Encodings like TensoRF or K-Planes rely on similar multilinear interpolation on axis-aligned structures. Therefore, they inherently suffer from the same grid-induced anisotropy analyzed in our work. The R-MHE concept (applying distinct rotations per level or per plane) could be directly applied to these architectures to improve isotropy. We have added this discussion to the Conclusion.

---

> ### Author Response · Authors · 2025-11-28
>
> Dear Reviewer,
>
> I hope this message finds you well. Our article has been updated and revised significantly, and we hope that we addressed all of your concerns. Since the discussion period is concluding, if there is anything else you would like us to clarify, please let us know. Thank you for your time and efforts in reviewing our paper.

---

### Official Review · Reviewer_6KVQ · 2025-10-31

**Soundness:** 3
**Presentation:** 3
**Contribution:** 3
**Rating:** 6
**Confidence:** 3

**Summary:**

## Summary
* This paper study the Multi Resolution Hash Encodings (MHE) proposed by instant-ngp, which is used to accelerate representation of 3D scene.
* This paper characterize the PFS of MHE in collision free assumption, concluding both theoretically and empirically that the effective resolution of MHE is not controled by max resolution $N_{MAX}$ as described by instant-ngp, but by average resolution $N_{AVG}$.
* Besides, this paper makes two practical contribution to HME when considering collision: a scheme for parameter selection and an improvement named R-MHE.

**Strengths:**

## Strength
* The characterization of MHE using PFS is novel and interesting. The conclusion that FWHM is direction dependent and its width is determined by $N_{max}$ is a very useful insight that might inspire later works.
* The empirical on 2D Image Regression is limited but convincing.

**Weaknesses:**

## Weakness
* The proposed approach is said to have two advantages, avoiding anisotropy and avoiding hashing collision. Both of the two advantages improve the empirical result. However, either because it is not possible to study the two factors independently, or the authors have not done it, I find it hard to evaluate the contribution of avoiding anisotropy to practical results. In fact, the majority of this paper is about how MHE introduces anisotropy. And I am definitely more interested in how the anisotropy property harms the performance of MHE beyond toy examples. While the current empirical result is not strong enough to show it.
* The MHE in its original paper (instant-ngp) is well evaluated with different tasks and contains a lot more formal comparsion with other methods. I do agree that this paper has its unique theoretical insights. However, implemented their proposed approach for other tasks and include more comparsion strengthen this paper a lot.

**Questions:**

* Is it possible to study the effect of anisotropy reduction and collision avoidance? Which one contributes mainly to the gain in PSNR?

---

> ### Author Response · Authors · 2025-11-23
>
> We appreciate the reviewer recognizing the novelty of the PSF characterization and the useful insight that FWHM is determined by $N_{avg}$.
>
> W1 & Q1: Disentangling contribution of anisotropy reduction and collision avoidance. Which one contributes mainly to the gain?
>
> Thank you for this important point. In our initial submission, the R-MHE architecture involved averaging multiple independent hash tables, which simultaneously addressed anisotropy (via rotation) and collision noise (via independent hashing and averaging). This made it difficult to disentangle the two effects, and the additional hashing would result in computational overhead, which was observed during the training process
>
> We have significantly revised the R-MHE architecture (Section 5). The new R-MHE applies rotations across different *levels* within a *single* MHE structure. This revised approach specifically isolates the benefit of **improved isotropy**, as it does not alter the collision statistics compared to the baseline MHE. While this means the revised R-MHE does not benefit from collision noise averaging (unlike the initial submission's design), it ensures a fair comparison and confirms that the performance gains demonstrated in the revised manuscript are directly attributable to the reduction of anisotropy.
>
> W1 (cont.): How anisotropy harms performance beyond toy examples.
>
> The 2D image regression results (Section 5.3, Fig. 5c-e) provide strong evidence. Standard MHE exhibits artifacts on textures not aligned with the axes. R-MHE mitigates these artifacts, leading to a substantial improvement in PSNR (Table 1). This demonstrates the negative impact of anisotropy on real-world signals. Furthermore, the consistent improvements in 3D NeRF demonstrate the practical benefits of mitigating anisotropy.
>
> W2: Limited tasks and comparisons.
>
> Our primary contribution is the novel theoretical framework (PSF analysis) and the R-MHE modification. To strengthen the empirical validation, we have expanded our experiments. The revision now includes results on 2D image regression, all eight scenes of the 3D Synthetic NeRF dataset, and a new task: 3D Neural SDF representation (Section 6). We believe this comprehensive evaluation demonstrates the broad applicability of our insights.

---

> ### Author Response · Authors · 2025-11-28
>
> Dear Reviewer,
>
> I hope this message finds you well. Our article has been updated and revised significantly, and we hope that we addressed all of your concerns. Since the discussion period is concluding, if there is anything else you would like us to clarify, please let us know. Thank you for your time and efforts in reviewing our paper.

---

### Official Review · Reviewer_Rkrg · 2025-11-01

**Soundness:** 3
**Presentation:** 3
**Contribution:** 3
**Rating:** 6
**Confidence:** 3

**Summary:**

This paper presents to provide understandings on multi-resolution hash encoding (MHE) originally presented from Instant-NGP.

In summary, the paper:
- Frames MHE as a physical system via its point spread function (PSF); derives a closed-form showing logarithmic decay and grid-induced anisotropy.
- Shows effective resolution follows the broadened empirical Full-width half-maximum (FWHM) = 1/N_avg (instead of N_max) due to optimization-induced broadening (beta=3).
- Analyzes finite-capacity collisions causing speckle noise and SNR loss; proposes Rotated MHE (R-MHE) with independently hashed, rotated grids to improve isotropy and SNR.
The paper backs its claims with experimental results, through 2D and 3D evaluations.

**Strengths:**

The paper provides interesting extension of multi-resolution hash encoding. It provides:
- Clear theory–practice link: closed-form PSF, anisotropy factor, and FWHM = 1/N_avg give actionable design rules.
- Comprehensive empirical validation: broadened PSF (beta=3) consistently matches measurements; two-point resolution aligns with FWHM, not N_max.
- Practical payoff with fixed memory: R-MHE markedly improves isotropy and PSNR in 2D (+8.06 dB to 31.30 dB at M=8) and yields steady 3D gains up to +0.65 dB.

**Weaknesses:**

Despite its strong theoretical foundations, the paper contains following potential shortcomings.
- Broadening factor of beta = 3 is largely empirical; sensitivity to optimizer/architecture/data is not exhaustively characterized in additional experiments.
- PSF-guided hyperparameter (b) underperforms a tuned baseline in 3D, and R-MHE’s 3D gains are modest (+0.65 dB).
- The paper fundamentally relies on MHE, of which multiple extensions such as dictionary fields has been proposed already. However, the paper does not compare with these extensions which limits its interpretability.

**Questions:**

1. How are the hyperparameters calculated? Where does beta value come from?
2. Why does performance gap between 2D and 3D MHE exist?

---

> ### Author Response · Authors · 2025-11-23
>
> We thank the reviewer for their positive assessment and constructive feedback.
>
> W1 & Q1: Broadening factor ($\beta$=3) is largely empirical; sensitivity not characterized. Where does $\beta$ come from?
>
> Thank you for highlighting this. We included a theoretical derivation (Appendix D.1-D.3) showing that the optimization-induced broadening ( $\beta_{\text {opt }}$ ) arises from the spectral bias of gradient-based optimization, where lower frequencies (coarser grids) are prioritized. The total empirical broadening is $\beta_{e m p}=\beta_{\text {ideal }} \cdot \beta_{\text {opt }}$.
>
> In the revision, we summarized this theory in Section 3.2 and conducted new experiments (Appendix D.4) to characterize $\beta$ 's sensitivity. We found that $\beta$ is dependent on the optimizer choice (e.g., $\beta_{e m p} \approx 3.0$ for Adam variants, lower for others like SGD). For a given optimizer, $\beta$ is remarkably stable across different MHE configurations (L, b) and insensitive to MLP architecture. This confirms that $\beta$ is a robust characteristic of the optimization dynamics for a given setup.
>
> W2: PSF-guided hyperparameter (b) underperforms a tuned baseline in 3D; modest 3D gains.
>
> Regarding the hyperparameter (b), the "Baseline Heuristic" default was already extremely close to the theoretically derived optimum, possibly obtained by grid search. The theory validates the heuristic in this specific scenario. The value of our approach is providing a principled way to determine the optimum without grid search, as demonstrated in our 2D/3D experiments.
>
> Regarding the 3D gains, we acknowledge they are modest (+0.13 dB in NeRF, Table 2). It is important to note that the larger gains reported in the initial submission (+0.65 dB) were achieved with the original R-MHE architecture, which benefited from both improved isotropy and collision noise averaging (by using multiple independent hash tables with significant extra computational costs). The revised R-MHE architecture isolates the effect of isotropy and provides this improvement at absolutely no extra cost (exact same parameters and computation). We believe a consistent, zero-overhead gain through a theoretically motivated modification is significant. We also added a new 3D SDF task (Section 6.3), where we find that performance saturates for both MHE and R-MHE, demonstrating that R-MHE maintains the high performance of the baseline in this regime.
>
> W3: Lack of comparison with MHE extensions (e.g., Dictionary Fields).
>
> We have updated the Related Work (Section 2.1) to discuss methods like Dictionary Fields. We emphasize our work is orthogonal; R-MHE improves the underlying grid structure and isotropy, and could potentially be combined with these approaches.
>
> Q2: Performance gap between 2D and 3D MHE.
>
> The larger relative gain in 2D likely stems from the task differences. The 2D experiments involve high-resolution images, placing greater demands on the encoding's isotropy; artifacts from anisotropy are often more pronounced here than in standard 3D NeRF datasets.

---

> ### Author Response · Authors · 2025-11-28
>
> Dear Reviewer,
>
> I hope this message finds you well. Our article has been updated and revised significantly, and we hope that we addressed all of your concerns. Since the discussion period is concluding, if there is anything else you would like us to clarify, please let us know. Thank you for your time and efforts in reviewing our paper.

---

### Official Review · Reviewer_4sPL · 2025-11-05

**Soundness:** 3
**Presentation:** 1
**Contribution:** 2
**Rating:** 4
**Confidence:** 3

**Summary:**

This paper proposes rotated multi-resolution hash encodings, widely used for implicit neural representations in image regression and neural radiance fields. Previous hash encodings are dependent on axes, leading to anisotropy and hindering empirical performance. They fail to capture well frequency patterns like sinusoidal functions because hash encodings are implicitly aligned with the axes of Cartesian coordinates. In contrast, this paper proposes rotated multi-resolution hash encodings that allow us to be independent of Cartesian coordinates. It achieves this by using multiple rotation functions for input position $x$, resulting in an average of randomly sampled rotation functions. This approach mitigates the reliance on representation aligned with Cartesian coordinates and improves prediction performance without degradation in frequency-based patterns. One of the randomly sampled rotation functions may potentially match the directions of the frequency patterns. This paper demonstrates its effectiveness in theoretical and practical cases.

**Strengths:**

- This paper introduces a novel approach that explicitly uses multiple random rotation transforms. To maintain the total number of parameters, the paper splits the original feature dimensions into the number of rotation and new feature dimensions. Although it employs a smaller number of feature dimensions, the random rotation function for input $x$ enables it to effectively capture signals that do not align with Cartesian coordinates.

- The paper demonstrates its effectiveness through both mathematical and practical cases. Specifically, it shows that the proposed approach consistently improves prediction performance in image regression and neural radiance fields, while maintaining the same total number of parameters across each case.

**Weaknesses:**

- I believe this paper falls short of meeting the standards of top-tier AI conferences like ICLR. The presentation lacks an effective and efficient way for readers to grasp the main contribution of the paper. While the mathematical contributions are included in the Appendix, and very abstracted equations are elaborated in the manuscript, I disagree that all the content in the manuscript must be abstracted. The Appendix is necessary to fully digest the mathematical details, but it doesn’t justify the need for the entire manuscript to be abstracted. I believe important equations (Eq 2 and 3) and their impact (e.g., FWMH) should be provided in the manuscript, but they aren’t.

- Empirically, maintaining the total number of parameters is crucial for improving prediction performance. However, this paper artificially designs baselines and their parameterization. For instance, its baselines started with a feature dimension of 8, which is not the official parameterization of feature dimensions in NeRFs, which sets it to 2 (To my best knowledge). Therefore, it lacks the space to adjust and accommodate random rotation functions. In this case, the proposed approach naturally faces challenges in its practical implementation.

- Following the second concern in Weakness, they should identify new cases where high-dimensional encodings are required to achieve accurate prediction performance. In that sense, they must demonstrate that rotated MHE not only maintains performance but also reduces the total number of parameters.

**Questions:**

- I’m curious about Figures 1, 2, and 3. The colors in these figures indicate different parameterizations and the value seems to be continuous in all of them. However, the color information is actually discrete because of the parameterization used in this experiment. Adopting a legend for continuous values makes it challenging to understand graphical explanations easily.

- Figure 1 and 2 are particularly difficult to grasp the main messages from. They require more detailed explanations and concepts to understand.

---

> ### Author Response · Authors · 2025-11-23
>
> We sincerely thank the reviewer for the detailed feedback, which has greatly helped improve the manuscript, particularly the presentation.
>
> W1: Presentation Lacks Effectiveness; Important Equations (Eq 2 and 3) Missing.
>
> We agree that the previous abstraction hindered readability. We have substantially revised the manuscript to improve clarity and theoretical rigor:
>
> 1. We moved the key theoretical derivations (Eq. 2,3) from the Appendix directly into Section 3.1.
> 2. We revised Section 3 to better emphasize the impact of these equations. Furthermore, we have significantly enhanced the theoretical rigor by adopting the induced kernel (B-spline) framework, which characterizes the average spatial response. We introduced a new derivation (Appendix A.4) using Taylor expansion to rigorously characterize the grid-induced anisotropy.
> 3. We added a new conceptual visualization (Fig. 1) early in the paper to provide a clearer intuition of the MHE kernel analysis and the R-MHE architecture.
>
> W2: Baselines started with a feature dimension of 8, which is not the official parameterization... which sets it to 2.
>
> We appreciate the reviewer raising this crucial point. We have addressed this comprehensively by refining the R-MHE architecture and updating our experimental protocol.
>
> 1. **Revised R-MHE Architecture:** We revised R-MHE (Section 5). Instead of averaging multiple independent hash tables (which required starting at F=8 in the previous version), the new R-MHE applies distinct rotations *at each resolution level* within the existing MHE structure. This ensures that R-MHE has the exact same parameter count and computational efficiency as standard MHE.
> 2. **Standard Configuration (F=2):** All experiments (2D and 3D) have been rerun using the standard Instant-NGP configuration (L=16, F=2). The revised R-MHE demonstrates consistent improvements over the baseline in this standard regime (Tables 1, 2, 3).
>
> W3: They must demonstrate that rotated MHE not only maintains performance but also reduces the total number of parameters.
>
> The revised R-MHE architecture is designed to improve performance with the exact same number of parameters. The consistent gains observed (Tables 1, 2, 3) demonstrate that R-MHE is a more efficient parameterization, providing improvements at zero overhead.
>
> Q1 & Q2: Figures 1, 2, and 3 Clarity. Adopting a legend for continuous values makes it challenging.
>
> We apologize for the confusion caused by the continuous color bars representing discrete parameter variations ($L$ and $b$). We have completely overhauled these figures (now Figures 2, and 4). As the FWHM is inherently continuous we only retain the continuous colorbar in Figure 3. We now use discrete color legends clearly indicating the parameter values, and have expanded the corresponding text explanations and captions.

---

> ### Author Response · Authors · 2025-11-28
>
> Dear Reviewer,
>
> I hope this message finds you well. Our article has been updated and revised significantly, and we hope that we addressed all of your concerns. Since the discussion period is concluding, if there is anything else you would like us to clarify, please let us know. Thank you for your time and efforts in reviewing our paper.

---

### Author Response · Authors · 2025-11-23
**General Response to the Committee**

We sincerely thank all reviewers for their thoughtful and constructive feedback, which has significantly improved the quality and clarity of our manuscript. We are encouraged that the reviewers found our core contribution (the PSF analysis of MHE) to be novel, interesting (6KVQ), clearly modeled (QjpZ), and providing a clear theory-practice link (Rkrg).

We have conducted a thorough revision addressing the concerns raised. The main updates are summarized below:

1. **Strengthened Theoretical Foundation and Anisotropy Derivation:** We significantly enhanced the rigor of our theoretical framework (Section 3.1). We revised the analysis to utilize the induced kernel (B-spline) framework, which rigorously characterizes the spatial response averaged over all grid alignments (addressing QjpZ's concern about alignment assumptions). We introduced a new, rigorous derivation using Taylor expansion (Appendix A.4), formally proving that the kernel is inherently narrowest along the axes and monotonically broadens towards the diagonals. Furthermore, we introduce a generalized analysis (K-sparse directions) that quantifies the Anisotropy Ratio and proves that anisotropy increases with dimensionality.

2. **Architectural Refinement of R-MHE:** We refined the R-MHE architecture to address potential concerns about fair comparison and parameter efficiency. The revised R-MHE applies distinct rotations at each resolution level within the existing MHE structure, rather than averaging multiple independent hash tables. This ensures R-MHE has the exact same parameter count and computational cost as standard MHE. This architectural revision ensures a fair comparison (addressing 4sPL) and also isolates the effects of isotropy from potential collision averaging (addressing 6KVQ). This confirms that the observed improvements in the revised manuscript are solely due to mitigated anisotropy at zero overhead.

3. **Enhanced Empirical Robustness:** All experiments (2D Image Regression, 3D NeRF, and the new 3D SDF task) have been rerun using the standard Instant-NGP configuration (F=2) across 5 random seeds, and we now report the mean and standard deviation (addressing QjpZ, 4sPL).

4. **Clarification and Sensitivity Analysis of the Broadening Factor:** We improved the connection between the empirical broadening ($\beta_\text{emp}$) and its theoretical origins by decomposing it into the idealized kernel width ($\beta_\text{ideal}$) and optimization-induced broadening ($\beta_\text{opt}$) arising from spectral bias (Appendix D). We added new experiments (Section 3.2, Appendix D.4) characterizing $\beta$'s sensitivity to optimizer choice and stability across training (addressing Rkrg, QjpZ).

5. **Presentation Overhaul:** We significantly improved the presentation by integrating key equations (Eq. 2, 3) into the main text, adding a new conceptual figure (Fig. 1), and redesigning Figures 2-5 for clarity by using discrete legends and clearer labeling (addressing 4sPL).

6. **Validation of PSF-Guided Optimization:** We validated our theoretical framework by demonstrating that the PSF-guided hyperparameter selection (growth factor 'b'), incorporating the broadening factor $\beta_\text{emp}$, accurately predicts the empirical optimum across 2D and 3D tasks (e.g., Appendix Fig. 8), providing a principled alternative to grid search (addressing Rkrg, QjpZ).

Minor modifications include that we fixed a bug on the rectangular image being stretched for the 2D image regression task (now center-cropped to square, Appendix B.1) and we moved the discussion of MLP depth to the appendix (D.4).

We believe these revisions address the reviewers' concerns and strengthen the paper significantly. Detailed responses to individual points follow.

---

### Author Response · Authors · 2025-11-30
**Summary of Revisions and Rebuttals for the Area Chair**

Dear Area Chair,

We acknowledge the unprecedented situation regarding the OpenReview system and the resulting modifications to the ICLR review process. We appreciate your efforts in handling our submission under these challenging circumstances.

Given that reviewers are unable to engage in further discussion or update their assessments based on our rebuttal, we are providing this summary to assist in your evaluation of our manuscript, the initial reviews, and our detailed responses (posted Nov 23rd).

Our work introduces a novel Point Spread Function (PSF) analysis for Multi-Resolution Hash Encoding (MHE), providing a rigorous physical systems perspective. We were encouraged that reviewers found our approach **novel and interesting** (6KVQ), **clearly modeled** (QjpZ), and providing a **clear theory-practice link** (Rkrg).

The initial reviews raised valid concerns primarily regarding the presentation, the architecture of our proposed R-MHE, the rigor of the anisotropy derivation, and the empirical validation. We have conducted a substantial revision to address these points. We highlight the following key updates, which we believe resolve the main criticisms:

**1. Refinement of R-MHE Architecture (Addressing 4sPL, 6KVQ):**
*   *Initial Concerns:* Fairness and overhead of the original R-MHE design (which averaged independent hash tables), and the difficulty in disentangling the effects of isotropy vs. collision averaging (6KVQ).
*   **Revision:** We completely revised the R-MHE architecture (Section 5). It now applies distinct rotations *within* the existing MHE structure.
*   **Impact:** The revised R-MHE has the *exact same* parameter count and computational cost as standard MHE. This ensures a fair comparison (4sPL) and confirms that the observed performance gains are solely due to improved isotropy (6KVQ).

**2. Strengthened Theoretical Rigor and Anisotropy (Addressing QjpZ):**
*   *Initial Concerns:* Potential reliance on alignment assumptions in the theoretical framework (QjpZ) and the need for enhanced derivation of anisotropy.
*   **Revision:** We enhanced the theoretical foundation (Section 3.1) by adopting the induced kernel (B-spline) framework, which rigorously characterizes the spatial response averaged over all grid alignments. We added a new formal proof (Appendix A.4) using Taylor expansion to rigorously quantify the inherent grid-induced anisotropy.
*   **Impact:** The theoretical analysis is now rigorous and robust to alignment concerns.

**3. Enhanced Empirical Robustness and Standard Configurations (Addressing 4sPL, QjpZ):**
*   *Initial Concerns:* Requests for experiments using the standard Instant-NGP feature dimension (F=2) (4sPL) and concerns about the robustness of single-run results (QjpZ).
*   **Revision:** All experiments (2D Image Regression, 3D NeRF, and a new 3D SDF task) were rerun using the standard configuration (L=16, F=2). We now report the mean and standard deviation across 5 random seeds (Tables 1-3).
*   **Impact:** The revised manuscript demonstrates consistent, statistically significant improvements using the standard, efficient configuration at zero overhead.

**4. Clarification of the Broadening Factor $\beta$ (Addressing Rkrg, QjpZ):**
*   *Initial Concerns:* The broadening factor $\beta$ appeared largely empirical, and sensitivity analysis was requested.
*   **Revision:** We clarified the origins of $\beta$, decomposing it into the idealized width ($\beta_{ideal}$) and optimization-induced broadening ($\beta_{opt}$) arising from spectral bias (Section 3.2, Appendix D). We added new experiments characterizing $\beta$'s dependence on the optimizer and its stability across configurations.
*   **Impact:** We provide a theoretical justification for the observed broadening and validated that it is a robust characteristic of the optimization dynamics.

We believe our detailed responses and the significant revisions have addressed the reviewers' concerns, resulting in a much stronger and clearer manuscript. We thank you for your consideration during this challenging review process.

---

### Meta-Review · Area_Chair_Zqi1 · 2026-01-07

**Summary:**

The reviewers identified multiple weaknesses in the manuscript including too abstracted presentation (4sPL), artificial and potentially unfair baselines (4sPL), choice of the broadening factor (Rkrg, QjpZ), modest gains in some setups and missing/limited comparisons (Rkrg,6KVQ), missing evaluation of avoidance of anisotropy and hash collision independently (Rkrg), baseline consistently outperforming the theory in 3D (QjpZ). This paper has received four reviews: two marginally above the acceptance threshold, and two marginally below.

**Reviewer Concerns:**

In their rebuttal, the authors addressed most of the raised concerns, including proposing a major reworking of the presentation for the theoretical framework. Related to the limited tasks and comparisons, for the newly provided results the gain is not significant (within 1 std) and this should be clearly stated in the revised version of the manuscript, together with a discussion on limitations for the 3D case.

**Reviewer Scores:**

Overall, it is felt that the reviewers could have aligned to a marginally above acceptance threshold evaluation.

---

### Decision · Program_Chairs · 2026-01-26

Accept (Poster)